# EXPLAINING REPRESENTATION BOTTLENECKS OF CONVOLUTIONAL DECODER NETWORKS

## ABSTRACT

In this paper, we prove representation bottlenecks of a cascaded convolutional decoder[1] network, considering the capacity of representing different frequency components of an input sample. We conduct the discrete Fourier transform on each channel of the feature map in an intermediate layer of the decoder network. Then, we introduce the rule of the forward propagation of such intermediate-layer spectrum maps, which is equivalent to the forward propagation of feature maps through a convolutional layer. Based on this, we find that each frequency component in the spectrum map is forward propagated independently with other frequency components. Furthermore, we prove two bottlenecks in representing feature spectrums. First, we prove that the convolution operation, the zero-padding operation, and a set of other settings all make a convolutional decoder network more likely to weaken high-frequency components. Second, we prove that the upsampling operation generates a feature spectrum, in which strong signals repetitively appears at certain frequencies. We will release all codes when this paper is accepted.

## 1 INTRODUCTION

Deep neural networks (DNNs) have exhibited superior performance in many tasks. However, in recent years, many studies discovered some theoretical defects of DNNs, *e.g.*, the vulnerability to adversarial attacks (Goodfellow et al., 2014), and the difficulty of learning middle-complex interactions (Deng et al., 2022). Besides, other studies explained typical phenomena during the training of DNNs, *e.g.*, the double-descent phenomenon (Nakkiran et al., 2019), the information bottleneck hypothesis (Tishby & Zaslavsky, 2015), and the lottery ticket hypothesis (Frankle & Carbin, 2018).

In comparison, in this study, we propose a new perspective to investigate how a cascaded convolutional decoder[1] network represents features at different frequencies. *I.e.*, when we apply the discrete Fourier transform (DFT) to each channel of the feature map or the input sample, we try to prove which frequency components of each input channel is usually strengthened/weakened by the network. To this end, previous studies (Xu et al., 2019a; Rahaman et al., 2019) claimed that DNNs were less likely to encode high-frequency components. However, these studies focused on a specific frequency that took the landscape of the loss function on all input samples as the time domain. **In comparison, we focus on a fully different type of frequency**, *i.e.*, the frequency *w.r.t.* the DFT on an input image or a feature map.

• Reformulating forward propagation in the frequency domain. As the basis for subsequent theoretical proof, we discover that we can reformulate the traditional forward propagation of feature maps as a new forward propagation on the feature spectrum. We derive the rule that forward propagates spectrums of different channels through a cascaded convolutional network, which is mathematically equivalent to the forward propagation on feature maps through a cascaded convolutional network.

• Based on the reformulation of the forward propagation, we prove the following conclusions.

(1) *The layerwise forward propagation of each frequency component of the spectrum map is independent with other frequency components.* In the forward propagation process, each frequency component of the feature spectrum is forward propagated independently with other frequency components, if the convolution operation does not change the size of the feature map in each channel. In

---

[1] Here, the decoder represents a typical network, whose feature map size is non-decreasing during the forward propagation.

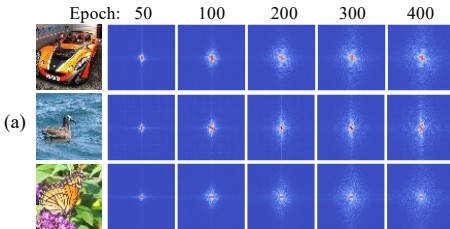 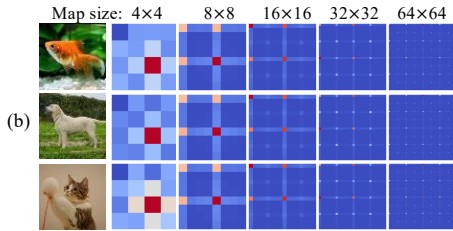

Figure 1: Two representation bottlenecks of a cascaded convolutional decoder network. (a) The convolution operation and the zero-padding operation make the decoder usually learn low-frequency components first and then gradually learn higher frequencies. (b) For cascaded upconvolutional layers, the upsampling operation in the decoder repeats strong frequency components of the input to generate spectrums of upper layers. We visualize the magnitude map of the feature spectrum, which is averaged over all channels. For clarity, we move low frequencies to the center of the spectrum map, and move high frequencies to corners of the spectrum map. High frequency components in magnitude maps in (b) are also weakened by the convolution operation after upsampling.

this way, we analyze three classic operations, including the convolution, the zero-padding, and the upsampling operations, and prove two representation bottlenecks, as follows.

(2) *Representation bottleneck 1.* We prove that both the convolution operation and the zero-padding operation make a cascaded convolutional decoder network more likely to weaken the high-frequency components of the input sample, as shown in Figure 1(a), if the convolution operation with a padding operation does not change the size of the feature map in a channel.

Besides, we also prove that the following three conditions further strengthen the above representation bottleneck, including (1) a deep network architecture; (2) a small convolutional kernel size; and (3) a large absolute value of the mean value of convolutional weights.

(3) *Representation bottleneck 2.* We porve that the upsampling operation makes a cascaded convolutional decoder network generate a feature spectrum, in which strong signals repetitively appears at certain frequencies, as shown in Figure 1(b).

Note that all above findings can explain general trends of neural networks with convolution, zero-padding, and upsampling operations, instead of deriving the deterministic property of a specific network. Besides, we have not derived the property of max-pooling operations, so in this paper, it is difficult to extend such findings to neural networks for image classification.

## 2 RULES OF PROPAGATING FEATURE SPECTRUMS

In this section, we aim to reformulate the forward propagation of a cascaded convolutional decoder[1] network in the frequency domain. To this end, we first introduce the rule of a convolutional layer propagating the feature spectrum from a lower layer to an upper layer.

● **Convolution operation.** Given a convolutional layer, let $\mathbf{W}^{[ker=1]}$, $\mathbf{W}^{[ker=2]}, \ldots, \mathbf{W}^{[ker=D]}$ denote $D$ convolutional kernels of this layer, and let $b^{[ker=1]}, b^{[ker=2]}, \ldots, b^{[ker=D]} \in \mathbb{R}$ denote $D$ bias terms. Each $d$-th kernel $\mathbf{W}^{[ker=d]} \in \mathbb{R}^{C \times K \times K}$ is of the kernel size $K \times K$, and $C$ denotes the channel number. Accordingly, we apply the kernel on a feature $\mathbf{F} \in \mathbb{R}^{C \times M \times N}$ with $C$ channels, and obtain the output feature $\widetilde{\mathbf{F}} \in \mathbb{R}^{D \times M' \times N'}$, as follows.

$$\widetilde{\mathbf{F}} = Conv(\mathbf{F}), \qquad \text{s.t. } \widetilde{\mathbf{F}}^{(d)} = \mathbf{W}^{[ker=d]} \otimes \mathbf{F} + b^{[ker=d]}\mathbf{1}_D, \ d = 1, 2, \ldots, D \qquad (1)$$

where $\widetilde{\mathbf{F}}^{(d)} \in \mathbb{R}^{M' \times N'}$ denotes the feature map of the $d$-th channel. $\otimes$ denotes the convolution operation. $\mathbf{1}_D \in \mathbb{R}^D$ is an all-ones vector.

● **Discrete Fourier transform.** Given the $c$-th channel of the feature $\mathbf{F} \in \mathbb{R}^{C \times M \times N}$, *i.e.*, $F^{(c)} \in \mathbb{R}^{M \times N}$, we use the discrete Fourier transform (DFT) (Sundararajan, 2001) to compute the frequency spectrum of this channel, which is termed $G^{(c)} \in \mathbb{C}^{M \times N}$, as follows. $\mathbb{C}$ denotes the algebra of complex numbers.

$$G_{uv}^{(c)} = \sum_{m=0}^{M-1} \sum_{n=0}^{N-1} F_{mn}^{(c)} e^{-i(\frac{um}{M} + \frac{vn}{N})2\pi}, \qquad u = 0, \ldots, M-1; \ v = 0, \ldots, N-1 \qquad (2)$$

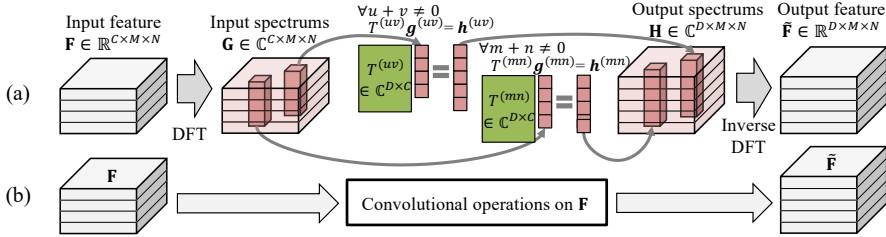

Figure 2: (a) Forward propagation in the frequency domain and (b) forward propagation in the time domain. The convolution operation on an input feature **F** is essentially equivalent to matrix multiplication on spectrums **G** of the feature.

Each frequency component at the frequency $[u, v]$ is represented as a complex number, *i.e.*, $G_{uv}^{(c)} \in \mathbb{C}$. Let $\mathbf{G} = [G^{(1)}, \ldots, G^{(C)}] \in \mathbb{C}^{C \times M \times N}$ denote the tensor of frequency spectrums of the $C$ channels of **F**. We take the $C$-dimensional vector at the frequency $[u, v]$ of the tensor **G**, *i.e.*, $\mathbf{g}^{(uv)} = [G_{uv}^{(1)}, G_{uv}^{(2)}, \ldots, G_{uv}^{(C)}]^\top \in \mathbb{C}^C$, to represent the frequency component $[u, v]$. Frequency components closed to $[0, 0], [0, N-1], [M-1, 0]$, or $[M-1, N-1]$ represent low-frequency signals, whereas frequency components closed to $[M/2, N/2]$ represent high-frequency signals.

• **Reformulating the *layerwise* forward propagation process.** For a specific convolutional layer (the stride size of the convolution operation is 1), the rule of propagating spectrums of input features into spectrums of output features is given as follows, which well represents the traditional forward propagation of features in Equation (1).

*Theorem 1.* (proven in Appendix A.1). *Let* $\mathbf{H} = [H^{(1)}, H^{(2)}, \ldots, H^{(D)}] \in \mathbb{C}^{D \times M' \times N'}$ *denote spectrums of the output feature* $\widetilde{\mathbf{F}} \in \mathbb{R}^{D \times M' \times N'}$. *Then,* $\mathbf{H}$ *can be computed as follows.*

$$\mathbf{h}^{(u'v')} = \delta_{u'v'} MN\mathbf{b} + \sum_{u=0}^{M-1} \sum_{v=0}^{N-1} \alpha_{u'v'uv} \cdot T^{(uv)} \mathbf{g}^{(uv)}, \qquad s.t. \ \delta_{u'v'} = \begin{cases} 1, & u' = 0; v' = 0 \\ 0, & otherwise \end{cases}; \quad (3)$$

$$\alpha_{u'v'uv} = \frac{1}{MN} \frac{\sin((M-K)\lambda_{uu'}\pi)}{\sin(\lambda_{uu'}\pi)} \frac{\sin((N-K)\gamma_{vv'}\pi)}{\sin(\gamma_{vv'}\pi)} e^{i((M-K)\lambda_{uu'} + (N-K)\gamma_{vv'})\pi}; \quad (4)$$

*where* $\mathbf{h}^{(u'v')} = [H_{u'v'}^{(1)}, H_{u'v'}^{(2)}, \ldots, H_{u'v'}^{(D)}]^\top \in \mathbb{C}^D$; $\mathbf{b} = [b^{(1)}, b^{(2)}, \ldots, b^{(D)}]^\top \in \mathbb{R}^D$ *denotes the vector of bias terms;* $\alpha_{u'v'uv} \in \mathbb{C}$ *is a coefficient;* $\lambda_{uu'} = \frac{(u-u')M - u(K-1)}{M(M-K+1)}$, $\gamma_{vv'} = \frac{(v-v')N - v(K-1)}{N(N-K+1)}$. $T^{(uv)} \in \mathbb{C}^{D \times C}$ *is a matrix of complex numbers, which is exclusively determined by convolutional kernels* $\mathbf{W}^{[ker=1]}, \mathbf{W}^{[ker=2]}, \ldots, \mathbf{W}^{[ker=D]}$.

$$T_{dc}^{(uv)} = \sum_{t=0}^{K-1} \sum_{s=0}^{K-1} W_{cts}^{[ker=d]} e^{i(\frac{ut}{M} + \frac{vs}{N})2\pi}, \qquad d = 1, 2, \ldots, D; \ c = 1, 2, \ldots, C. \quad (5)$$

In Equation (3), the $T^{(uv)}$ term corresponds to the interference process (Beaver, 2018) in physics, and the $\alpha_{u'v'uv}$ term corresponds to the diffraction process. According to Equation (4), for most $[u', v']$, $[u, v]$, $|\alpha_{u'v'uv}|$ is close to 0. $|\alpha_{u'v'uv}|$ is relatively large only when $[u', v']$ is close to $[u, v]$.

We notice that in most real implementations, the convolution operation does not change the size of the feature map in a channel by applying the padding operation. Decoder networks usually use the upsampling operation to increase the dimension of features. Therefore, we limit our research to the scope of convolution operations without changing the size of the feature map in a channel. Thus, we propose the following assumption, which is used in all subsequential proofs.

*Assumption 1. To simplify subsequential proofs, we assume that before each convolution operation, there exists a circular padding operation (Londono, 1982), and set the stride size of the convolution operation to 1. The circular padding operation is used to extend the last row and the last column of the feature map in each channel, so as to avoid the convolution changing the size of the feature map.*

Thus, keeping the feature map size unchanged removes the diffraction process term $\alpha_{u'v'uv}$ in theory, and derives Theorem 2. In fact, because $|\alpha_{u'v'uv}|$ is small in most cases, the diffraction process is actually ignorable, even when the convolution operation changes the size of the feature map.

*Theorem 2.* (proven in Appendix A.2). *Based on Assumption 1, the layerwise dynamics of feature spectrums in the frequency domain can be simplified as follows.*

$$\mathbf{h}^{(uv)} = T^{(uv)} \mathbf{g}^{(uv)} + \delta_{uv} MN\mathbf{b} \quad (6)$$

*Understanding the convolution operation in the frequency domain.* As Figure 2 shows, Theorem 2 means that conducting the convolution operation on an input feature $\mathbf{F}$ is essentially equivalent to conducting matrix multiplication on spectrums of $\mathbf{F}$. For example, for all frequencies except for the fundamental frequency, we have the output spectrum $\mathbf{h}^{(uv)} = T^{(uv)}\mathbf{g}^{(uv)}$.

**(Conclusion 1)** Each frequency component of the feature spectrum is propagated independently with other frequencies, $\mathbf{h}^{(uv)} = T^{(uv)}\mathbf{g}^{(uv)}$, where $T^{(uv)}$ is exclusively determined by convolutional weights. Therefore, $\mathbf{g}^{(uv)}$ is propagated independently with other frequency components $\mathbf{g}^{(u'v')}$.

• **Reformulating the *entire* propagation process of a cascaded convolutional network.** To simplify the further proof, we temporarily investigate the spectrum propagation of a network with $L$ cascaded convolutional layers, but without activation functions. Nevertheless, we have conducted various experiments, and experimental results in Figure 3 show that all our theorems can well reflect properties of an ordinary cascaded convolutional network with ReLU layers.

Let a convolutional network contain $L$ cascaded convolutional layers. Each $l$-th layer contains $C_l$ convolutional kernels, $\mathbf{W}^{(l)[ker=1]}, \mathbf{W}^{(l)[ker=2]}, \ldots, \mathbf{W}^{(l)[ker=C_l]} \in \mathbb{R}^{C_{l-1} \times K \times K}$, with $C_l$ bias terms $b^{(l,1)}, b^{(l,2)}, \ldots, b^{(l,C_l)} \in \mathbb{R}$. Let $x \in \mathbb{R}^{C_0 \times M \times N}$ denote the input sample. The network generates the output sample $\widehat{x} = \text{net}(x) \in \mathbb{R}^{C_L \times M \times N}$. Then, we derive the forward propagation of spectrums of $x$ to spectrums of $\widehat{x}$ in the frequency domain as follows.

*Corollary* 1. (proven in Appendix A.3.) *Let* $\boldsymbol{G} = [G^{(1)}, G^{(2)}, \ldots, G^{(C_0)}] \in \mathbb{C}^{C_0 \times M \times N}$ *denote frequency spectrums of the $C_0$ channels of the input $x$. Then, based on Assumption 1, spectrums of the image $\widehat{x}$ generated by $L$ cascaded convolutional layers, i.e.,* $\boldsymbol{H} = [H^{(1)}, H^{(2)}, \ldots, H^{(C_L)}] \in \mathbb{C}^{C_L \times M \times N}$ *is given as*

$$\boldsymbol{h}^{(uv)} = \mathbb{T}^{(uv)(L:1)}\boldsymbol{g}^{(uv)} + \delta_{uv}\boldsymbol{\beta} \tag{7}$$

*where* $\mathbb{T}^{(uv)(L:1)} = T^{(L,uv)} \cdots T^{(2,uv)}T^{(1,uv)} \in \mathbb{C}^{C_L \times C_0}$; $\boldsymbol{g}^{(uv)} = [G_{uv}^{(1)}, G_{uv}^{(2)}, \ldots, G_{uv}^{(C_0)}]^\top \in \mathbb{C}^{C_0}$ *and* $\boldsymbol{h}^{(uv)} = [H_{uv}^{(1)}, H_{uv}^{(2)}, \ldots, H_{uv}^{(C_L)}]^\top \in \mathbb{C}^{C_L}$ *denote vectors at the frequency $[u,v]$ in tensors $\boldsymbol{G}$ and $\boldsymbol{H}$, respectively.* $\boldsymbol{\beta} = MN\big(\boldsymbol{b}^{(L)} + \sum_{j=2}^{L} \mathbb{T}^{(00)(L:j)}\boldsymbol{b}^{(j-1)}\big) \in \mathbb{C}^{C_L}$; $\boldsymbol{b}^{(l)} = [b^{(l,1)}, b^{(l,2)}, \ldots, b^{(l,C_l)}]^\top \in \mathbb{R}^{C_l}$ *denotes the vector of bias terms of $C_l$ convolutional kernels in the $l$-th layer.*

Besides, the learning of parameters $\mathbf{W}^{(l)}$ affects the matrix $T^{(l,uv)}$. Therefore, we further reformulate the change of $T^{(l,uv)}$ during the learning process, as follows.

*Corollary* 2. (proven in Appendix A.4.) *Based on Assumption 1, the change of each frequency components $T^{(l,uv)}$ during the learning process is reformulated as follows.*

$$\Delta T^{(l,uv)} = -\eta MN \sum_{u'=0}^{M-1}\sum_{v'=0}^{N-1} \chi_{u'v'uv}\left(\overline{\mathbb{T}}^{(u'v')(l-1:1)}\overline{\boldsymbol{g}}^{(u'v')} + \delta_{u'v'}\overline{\boldsymbol{\beta}}'\right)\frac{\partial Loss}{\partial(\overline{\boldsymbol{h}}^{(u'v')})^\top}\overline{\mathbb{T}}^{(u'v')(L:l+1)}; \tag{8}$$

$$s.t. \ \chi_{u'v'uv} = \frac{1}{MN}\frac{\sin(K(u-u')\pi/M)}{\sin((u-u')\pi/M)}\frac{\sin(K(v-v')\pi/N)}{\sin((v-v')\pi/N)}e^{i(\frac{(K-1)(u-u')}{M} + \frac{(K-1)(v-v')}{N})\pi} \tag{9}$$

*where* $\eta$ *is the learning rate;* $\chi_{u'v'uv} \in \mathbb{C}$ *is a coefficient;* $\mathbb{T}^{(u'v')(l-1:1)} = T^{(l-1,u'v')} \cdots T^{(2,u'v')}T^{(1,u'v')} \in \mathbb{C}^{C_{l-1} \times C_0}$; $\mathbb{T}^{(u'v')(L:l+1)} = T^{(L,u'v')} \cdots T^{(l+1,u'v')} \in \mathbb{C}^{C_L \times C_l}$; $\boldsymbol{\beta}' = MN\big(\boldsymbol{b}^{(l-1)} + \sum_{j=2}^{l-1} \mathbb{T}^{(00)(l-1:j)}\boldsymbol{b}^{(j-1)}\big) \in \mathbb{C}^{C_{l-1}}$; $\overline{\mathbb{T}}^{(uv)(l-1:1)}$ *is the conjugate of* $\mathbb{T}^{(uv)(l-1:1)}$.

**Verifying the forward propagation in Corollary 1 and the change of $T^{(l,uv)}$ in Corollary 2.** We computed the similarity between real spectrums $\mathbf{H}^* = [H^{*(1)}, H^{*(2)}, H^{*(3)}, \cdots]$ measured by applying the DFT to the real network output, and spectrums $\mathbf{H} = [H^{(1)}, H^{(2)}, H^{(3)}, \cdots]$ derived in Corollary 1, in order to verify the correctness of the forward propagation in the frequency domain. Specifically, we measured the cosine similarity $similarity(\mathbf{H}^*, \mathbf{H}) = \mathbb{E}_c[\mathbf{cos}(\mathbf{vec}(\mathbf{norm}(H^{*(c)})), \mathbf{vec}(\mathbf{norm}(H^{(c)})))]$, where $\mathbf{vec}(\cdot)$ represents the vectorization of a matrix, and $\mathbf{norm}(\cdot)$ represents computing the norm of each complex number in a matrix.

To this end, we constructed the following three baseline networks to verify whether Corollary 1 derived from specific assumptions could also objectively reflect real forward propagations in real neural networks. Specifically, the first baseline network contained 10 convolutional layers. Each convolutional layer applied zero-paddings and was followed by an ReLU layer. Each convolutional layer contained 16 convolutional kernels (kernel size was $3 \times 3$) with 16 bias terms. We set the stride size of the convolution operation to 1. The second baseline network was constructed by removing all ReLU layers from the first baseline network, which was closer to the assumption in Corollary 1.

Figure 3: (a) Fitness between the derived feature spectrums $\mathbf{H}$ in Corollary 1 and the real feature spectrums $\mathbf{H}^*$ measured in a real DNN. (b) Fitness between the derived change of $T^{(l,uv)}$ in Corollary 2 and the real $T^{(l,uv)}$ measured in a real DNN. The shaded area represents the standard deviation.

The third baseline network was constructed by replacing all zero-paddings with circular paddings from the second baseline network, which was exactly the same with the assumption in Corollary 1.

Figure 3(a) reports *similarity*($\mathbf{H}^*$, $\mathbf{H}$) that was measured on spectrums in different layers and averaged over all samples. The similarity between real spectrums and derived spectrums was large for all the three baseline networks, which verified Corollary 1. Note that the cosine similarity was computed based on high-dimensional vectors with as many as $32^2$ or $64^2$ or $224^2$ dimensions (determined by the dataset), in which case tiny noises were accumulated significantly. Therefore, the similarity greater than 0.8 was already significant enough to verify the practicality of our theory.

Besides, we also measured the similarity between the real change of $T^{(l,uv)}$ computed by measuring real network parameters, termed $\Delta^* T^{(l,uv)}$, and the change of $T^{(l,uv)}$ derived with certain assumptions in Corollary 2, *i.e.*, $\Delta T^{(l,uv)}$, in order to verify Corollary 2. The similarity was also computed as *similarity*($\Delta^* T^{(l,uv)}$, $\Delta T^{(l,uv)}$) $= \mathbb{E}_c[\mathbf{cos}(\mathbf{vec}(\mathbf{norm}(\Delta^* T^{(l,uv)})), \mathbf{vec}(\mathbf{norm}(\Delta T^{(l,uv)})))]$. The verification was also conducted on the above three baseline networks. Figure 3(b) reports $\forall l$, *similarity*($\Delta^* T^{(l,uv)}$, $\Delta T^{(l,uv)}$) averaged over all samples. The similarity was greater than 0.88 for all the three baseline networks, which verified Corollary 2.

## 3 REPRESENTATION BOTTLENECKS

We further analyze the effects of three classic operations on representing different frequency components of an input sample, including the convolution operation, the zero-padding operation, and the upsampling operation, and discover two representation bottlenecks.

• **Effects of the convolution operation.** Given an initialized, cascaded, convolutional decoder[1] network with $L$ convolutional layers, let us focus on the behavior of the decoder network in early epochs of training. We notice that each element in the matrix $T^{(l,uv)}$ is exclusively determined by the $c$-th channel of the $d$-th kenel $W_{c,0:K-1,0:K-1}^{(l)[ker=d]} \in \mathbb{R}^{K \times K}$ according to Equation (5). Because parameters in $W^{(l)}$ in the decoder network are set to random noises, we can consider that all elements in $T^{(l,uv)}$ irrelevant to each other, *i.e.*, $\forall d \neq d', c \neq c', T_{dc}^{(l,uv)}$ is irrelevant to $T_{d'c'}^{(l,uv)}$. Similarly, since different layers' parameters $W^{(l)}$ are irrelevant to each other in the initialized decoder network, we can consider that elements in different layers' $T^{(l,uv)}$ irrelevant to each other, *i.e.*, $\forall l \neq l'$, elements in $T^{(l,uv)}$ and elements in $T^{(l',uv)}$ are irrelevant to each other. Moreover, since the early training of a DNN mainly modifies a few parameters according to the lottery ticket hypothesis (Frankle & Carbin, 2018), we can still assume such irrelevant relationships in early epochs, as follows.

*Assumption 2.* (proven in Appendix A.5) *We assume that all elements in $T^{(l,uv)}$ are irrelevant to each other, and $\forall l \neq l'$, elements in $T^{(l,uv)}$ and $T^{(l',uv)}$ are irrelevant to each other in early epochs.*

$$\forall d \neq d'; \forall c \neq c', \quad \mathbb{E}_{W^{(l)}}[T_{dc}^{(l,uv)} T_{d'c'}^{(l,uv)}] = \mathbb{E}_{W^{(l)}}[T_{dc}^{(l,uv)}] \mathbb{E}_{W^{(l)}}[T_{d'c'}^{(l,uv)}] \tag{10}$$

$$\forall l, d, c, d', c', \quad \mathbb{E}_{W^{(l)}, \ldots, W^{(1)}}[T_{dc}^{(l,uv)} \mathbb{T}_{d'c'}^{(uv)(l-1:1)}] = \mathbb{E}_{W^{(l)}}[T_{dc}^{(l,uv)}] \mathbb{E}_{W^{(l-1)}, \ldots, W^{(1)}}[\mathbb{T}_{d'c'}^{(uv)(l-1:1)}] \tag{11}$$

*Besides, according to experimental experience, the mean value of all parameters in $W^{(l)}$ usually has a small bias during the training process, instead of being exactly zero. Therefore, let us assume that in early epochs, each parameter in $W^{(l)}$ is sampled from a Gaussian distribution $N(\mu_l, \sigma_l^2)$.*

According to $\mathbf{h}^{(uv)} = \mathbb{T}^{(uv)(L:1)} \mathbf{g}^{(uv)} + \delta_{uv} MN\mathbf{b}$ in Corollary 1, each frequency component $\mathbf{h}^{(uv)}$ of the output spectrum is exclusively determined by the component $\mathbf{g}^{(uv)}$ of the input sample and the matrix $\mathbb{T}^{(uv)(L:1)} = T^{(L,uv)} \cdots T^{(2,uv)} T^{(1,uv)}$, since $\delta_{uv} = 0$ on all frequencies other than the fundamental frequency. Therefore, **the magnitude of $\mathbb{T}^{(uv)(L:1)}$ reflects the strength of the network encoding this specific frequency component $\mathbf{g}^{(uv)}$.**

*Theorem* 3. (proven in Appendix A.5) *Based on Assumption 1 and Assumption 2, we can prove that* $T_{dc}^{(l,uv)}$ *follows a Gaussian distribution of complex numbers, as follows.*

$$\forall d, c \quad T_{dc}^{(l,uv)} \sim Complex\mathcal{N}(\hat{\mu} = \mu_l R_{uv}, \hat{\sigma}^2 = K^2 \sigma_l^2, r = \sigma_l^2 R_{2u,2v}) \tag{12}$$

$$s.t. \quad R_{uv} = \frac{\sin(uK\pi/M)}{\sin(u\pi/M)} \frac{\sin(vK\pi/N)}{\sin(v\pi/N)} e^{i(\frac{(K-1)u}{M} + \frac{(K-1)v}{N})\pi} \tag{13}$$

*Different from the Gaussian distribution of real numbers, the Gaussian distribution of complex numbers has three parameters $\hat{\mu} \in \mathbb{C}, \hat{\sigma}^2 \in \mathbb{R}$ and $r \in \mathbb{C}$, which control the mean value, the variance, and the diversity of the phase of the sampled complex number, respectively. Specifically, a large value of $|r|$ indicates that the sampled complex number $T_{dc}^{(l,uv)}$ is less likely to have diverse phases. $R_{uv} \in \mathbb{C}$ is a complex coefficient, $0 \leq |R_{uv}| \leq K^2$.*

For a low-frequency component $[u^{\text{low}}, v^{\text{low}}]$, $|R_{u^{\text{low}}v^{\text{low}}}|$ is relatively large. Therefore, the second-order moment of $T_{dc}^{(l,u^{\text{low}}v^{\text{low}})}$, *i.e.*, $|\mu_l R_{u^{\text{low}}v^{\text{low}}}|^2 + K^2 R_{u^{\text{low}}v^{\text{low}}}^2$, is large, which indicates that the sampled $T_{dc}^{(l,u^{\text{low}}v^{\text{low}})}$ is more likely to have a large norm. Besides, the parameter $|r| = |\sigma_l^2 R_{2u^{\text{low}},2v^{\text{low}}}|$ is large for low frequencies, which means that the sampled $T_{dc}^{(l,u^{\text{low}}v^{\text{low}})}$ is less likely to have diverse phases. In contrast, for a high-frequency component $[u^{\text{high}}, v^{\text{high}}]$, the sampled $T_{dc}^{(l,u^{\text{high}}v^{\text{high}})}$ is less likely to have a large norm and is more likely to have diverse phases.

*Theorem* 4. (proven in Appendix A.6) *For the simplest case that each convolutional layer only contains a feature map with a single channel, i.e., $\forall l, C_l = 1$. Then, based on Theorem 3 and Assumption 2, $\mathbb{T}^{(uv)(L:1)} = T^{(L,uv)} \cdots T^{(2,uv)} T^{(1,uv)} \in \mathbb{C}$ follows a distribution, which is the product of L complex numbers, where each complex number follows a Gaussian distribution. The mean value of $\mathbb{T}^{(uv)(L:1)}$ is $\prod_{l=1}^{L} \mu_l R_{uv} \in \mathbb{C}$. The logarithm of the second-order moment is given as $\log SOM(\mathbb{T}^{(uv)(L:1)}) = \sum_{l=1}^{L} \log(|\mu_l R_{uv}|^2 + K^2 \sigma_l^2) \in \mathbb{R}$.*

For the more general case that each convolutional kernel contains more than one channel, *i.e.*, $\forall l, C_l > 1$, the $SOM(\mathbb{T}^{(uv)(L:1)})$ also approximately exponentially increases along with the depth of the network with a quite complicated analytic solution. Please see Appendix A.6 for the proof.

**(Conclusion 2)** Therefore, according to above proof, the convolution operation makes a cascaded convolutional decoder network more likely to weaken the high-frequency components of the input sample, if the convolution operation does not change the feature map size. Specifically, we obtain the following five remarks to specify detailed mechanisms of weakening high-frequency components.

*Remark* 1. According to Theorem 4, for each frequency component $[u, v]$, the second-order moment $SOM(\mathbb{T}^{(uv)(L:1)})$ will exponentially increase along with the depth $L$ of the network. **We can consider that each layer' $T^{(l,uv)}$ has independent effects** $\log(|\mu_l R_{uv}|^2 + K^2 \sigma_l^2)$ **on** $\log SOM(\mathbb{T}^{(uv)(L:1)}) = \sum_{l=1}^{L} \log(|\mu_l R_{uv}|^2 + K^2 \sigma_l^2)$.

We admit that the conclusion in Remark 1 is derived from the second-order moment of $\mathbb{T}^{(uv)(L:1)}$, instead of a deterministic claim for a specific neural network. Nevertheless, according to the Law of Large Numbers, $SOM(\mathbb{T}^{(uv)(L:1)})$ is still a convincing metric to reflect the significance of $\mathbb{T}^{(uv)(L:1)}$.

*Remark* 2. If the decoder network is deep, then the decoder network is less likely to learn high-frequency components. It is because $|R_{uv}|$ is relatively large for low-frequency components. In this way, the large effect of a single layer's $T^{(l,uv)}$ of low-frequency components on $\log SOM(\mathbb{T}^{(uv)(L:1)})$, *i.e.*, $\log(|\mu_l R_{uv}|^2 + K^2 \sigma_l^2)$, can be accumulated through different layers according to the Law of Large Numbers and the independence between different layers in Remark 1.

Therefore, the large $|R_{u^{\text{low}}v^{\text{low}}}|$ value for a low-frequency component $[u^{\text{low}}, v^{\text{low}}]$ makes $\mathbb{T}^{(u^{\text{low}}v^{\text{low}})(L:1)}$ more likely to have a large norm, whereas the small $|R_{u^{\text{high}}v^{\text{high}}}|$ value for a high-frequency component $[u^{\text{high}}, v^{\text{high}}]$ makes $\mathbb{T}^{(u^{\text{high}}v^{\text{high}})(L:1)}$ less likely to have a large norm. This indicates that **a deep decoder network will almost certainly strengthen the encoding of low-frequency components of the input sample, while weaken the encoding of high-frequency components.**

*Remark* 3. If the expectation $\mu_l$ of convolutional weights in each $l$-th layer has a large absolute value $|\mu_l|$, then the decoder network is less likely to learn high-frequency components. It is because according to Theorem 4, a large absolute value $|\mu_l|$ boosts the imbalance effects $|\mu_l R_{uv}|^2$ among different frequency components, thereby strengthening the trend of encoding low-frequency components of the input sample.

*Remark* 4. If the convolutional kernel size $K$ is small, then the decoder network is less likely to learn high-frequency components. It is because according to Theorem 4, a large $K$ value alleviates imbalance of the second-order moment $SOM(\mathbb{T}^{(uv)(L:1)})$ between low frequencies and high frequencies caused by the imbalance of $|R_{uv}|$. Thus, a small $K$ value strengthens the trend of encoding low-frequency components of the input sample.

*Remark* 5. If the cascaded convolutional decoder network is trained on natural images, then the decoder network is less likely to learn high-frequency components. Previous studies (Ruderman, 1994) have empirically found that natural images were dominated by low-frequency components. Specifically, frequency spectrums of natural images follow a Power-law distribution. *I.e.*, low-frequency components (*e.g.*, the frequency component $[u, v]$ closed to $[0, 0], [0, N-1], [M-1, 0]$, and $[M-1, N-1]$) have much larger length $\|\mathbf{g}^{(uv)}\|_2 = \sqrt{\sum_c |G_{uv}^{(c)}|^2}$ than other frequency components. Besides, according to rules of the forward propagation in Equation (7) and the change of $T^{(l, uv)}$ in Equation (8), if the frequency component $\mathbf{g}^{(uv)}$ of the input image has a large magnitude, then $\mathbf{h}^{(uv)}$ of the output image also has a large magnitude. This means that using natural images as the input strengthens the trend of encoding low-frequency components.

These five remarks tell us different ways to strengthen or weaken the capacity of a decoder of modeling specific frequency components. Experiments in Section 4 have verified Remarks 1 to 4 in the general case that each convolutional layer contains more than one feature maps.

● **Effects of the zero-padding operation.** To simplify the proof, let us consider the following one-size zero-padding. Given each $c$-th channel $F^{(c)} \in \mathbb{R}^{M \times N}$ of the feature map, the zero-padding puts zero values at the edge of $F^{(c)}$, so as to obtain a new feature $\widetilde{F}^{(c)} \in \mathbb{R}^{M' \times N'}$, as follows.

$$\forall m, n, \quad \widetilde{F}_{mn}^{(c)} = \begin{cases} F_{mn}^{(c)}, & 0 \le m < M,\ 0 \le n < N \\ 0, & M \le m < M',\ N \le n < N' \end{cases} \tag{14}$$

We have proven that the zero-padding operation boosts magnitudes of low-frequency components of feature spectrums of the feature map, as shown in Theorem 5.

*Theorem* 5. (proven in Appendix A.7) *Let each element in each $c$-th channel $F^{(c)}$ of the feature map follows the Gaussian distribution $\mathcal{N}(a, \sigma^2)$. $G^{(c)} \in \mathbb{C}^{M \times N}$ denotes the frequency spectrum of $F^{(c)}$, and $H^{(c)} \in \mathbb{C}^{M' \times N'}$ denotes the frequency spectrum of the output feature $\widetilde{F}^{(c)}$ after applying zero-padding on $F^{(c)}$. Then, the zero-padding on $F^{(c)}$ brings in additional signals at each frequency $[u, v]$ as follows, whose strength is measured by averaging over different sampled features.*

$$\forall 0 \le u < M,\ 0 \le v < N,\ \mathbb{E}_{F^{(c)}}[|H_{uv}^{(c)} - G_{uv}^{(c)}|] = |a| \left( \left| \frac{\sin(Mu\pi/M')}{\sin(u\pi/M')} \frac{Nv\pi/N'}{v\pi/N'} \right| - MN\delta_{uv} \right);$$

$$\forall M \le u < M',\ N \le v < N',\ \mathbb{E}_{F^{(c)}}[|H_{uv}^{(c)}|] = \left| a \frac{\sin(Mu\pi/M')}{\sin(u\pi/M')} \frac{Nv\pi/N'}{v\pi/N'} e^{-i(\frac{(M-1)u}{M'} + \frac{(N-1)v}{N'})\pi} \right| \tag{15}$$

**(Conclusion 3)** According to rules of the forward propagation in Equation (7) and the change of $T^{(l, uv)}$ in Equation (8), the zero-padding operation strengthens the trend of encoding low-frequency components of the input sample, because $\mathbb{E}_{F^{(c)}}[|H_{uv}^{(c)} - G_{uv}^{(c)}|]$ is large for low frequencies $[u, v]$.

● **Effects of the upsampling operation.** Let the $l$-th intermediate-layer feature map $\mathbf{F} \in \mathbb{R}^{C_l \times M_0 \times N_0}$ pass through an upsampling layer to extend its width and height to $M \times N$, subject to $M = M_0 \cdot ratio$, $N = N_0 \cdot ratio$ as follows.

$$\forall c, m^*, n^*,\ \widetilde{F}_{m^*n^*}^{(c)} = \begin{cases} F_{mn}^{(c)}, & \mathrm{mod}(m^*, ratio) = 0; \mathrm{mod}(n^*, ratio) = 0 \\ 0, & otherwise \end{cases} \quad s.t. \begin{cases} m = m^*/ratio \\ n = n^*/ratio \end{cases} \tag{16}$$

*Theorem* 6. (proven in Appendix A.8) *Let $\mathbf{G} = [G^{(1)}, G^{(2)}, \dots, G^{(C_l)}] \in \mathbb{C}^{C_l \times M_0 \times N_0}$ denote spectrums of the $C_l$ channels of feature $\mathbf{F}$. Then, spectrums $\mathbf{H} = [H^{(1)}, H^{(2)}, \dots, H^{(C_l)}] \in \mathbb{C}^{C_l \times M \times N}$ of the output feature $\widetilde{\mathbf{F}}$ can be computed as follows.*

$$\forall c, u, v,\ H_{u+(s-1)M_0, v+(t-1)N_0}^{(c)} = G_{uv}^{(c)} \quad s.t.\ s = 1, \dots, M/M_0;\ t = 1, \dots, N/N_0 \tag{17}$$

Theorem 6 shows that the upsampling operation repeats the strong magnitude of the fundamental frequency $G_{00}^{(c)}$ of the lower layer to different frequency components $\forall c, H_{u^*v^*}^{(c)}$ of the higher layer, where $u^* = 0, M_0, 2M_0, \dots; v^* = 0, N_0, 2N_0, \dots$. Such a phenomenon is shown in Appendix C.2.

Figure 4: (a) The exponential increase of the second-order moment of feature spectrums, $SOM(\mathbf{h}^{(uv)})$ (the linear increase of $\mathbf{log}SOM(\mathbf{h}^{(uv)})$ along with the layer number linearly). (b) A small kernel size $K$ usually made the network learn a higher proportion $p^{\text{low}}$ of low-frequency components.

**(Conclusion 4)** The upsampling operation makes the upconvolution operation generate a feature spectrum, in which strong signals of the input repetitively appears at certain frequencies. Such unexpected strong signals hurt the representation capacity of the network.

More crucially, according to the spectrum propagation in Corollary 1, such unexpected frequency components can be further propagated to upper layers. Thus, Corollary 1 may provide some clues to differentiate real samples and the generated samples.

## 4 EXPERIMENTS

• **Verifying that a neural network usually learned low-frequent components first.** Our theorems prove that a cascaded convolutional decoder network weakens the encoding of high-frequency components. In this experiment, we visualized spectrums of the image generated by a decoder network, which showed that the decoder usually learned low-frequency components in early epochs and then shifted its attention to high-frequency components. To this end, we constructed a cascaded convolutional auto-encoder by using the VGG-16 (Simonyan & Zisserman, 2015) as the encoder network. The decoder network contained four upconvolutional layers. Each convolutional/upconvolutional layer in the auto-encoder applied zero-paddings and was followed by a batch normalization layer and an ReLU layer. The auto-encoder was trained on the Tiny-ImageNet dataset (Le & Yang, 2015) using the mean squared error (MSE) loss for image reconstruction. Our theorem was verified by the well-known phenomenon in Figure 1(a), *i.e.*, an auto-encoder usually first generated images with low-frequency components, and then gradually generated more high-frequency components. In addition, Appendix C.1 shows results on more datasets, which also yielded similar conclusions.

• **Verifying that the upsampling operation made a decoder network repeat strong signals at certain frequencies of the generated image in Theorem 6.** To this end, we compared feature spectrums between the input spectrum and the output spectrum of the upsampling layer. We also conducted experiments on the auto-encoder introduced above. Figure 1(b) shows that the decoder network repeated strong signals at certain frequencies of the generated image. In addition, Appendix C.2 shows results on more datasets, which also yielded similar conclusions.

• **Verifying that the zero-padding operation strengthened the encoding of low-frequency components.** To this end, we compared feature spectrums between the network with zero-padding operations and the network without zero-padding operations. Therefore, we constructed the following two baseline networks. The first baseline network contained 5 convolutional layers, and each layer applied zero-paddings. Each convolutional layer contained 16 convolutional kernels (kernel size was $7 \times 7$), except for the last layer containing 3 convolutional kernels. The second baseline network was constructed by replacing all zero-padding operations with circular padding operations. Results on the Broden dataset in Figure 5(c) show that the network with zero-padding operations encoded more significant low-frequency components than the network with circular padding operations. In addition, Appendix C.3 shows results on more datasets, which also yielded similar conclusions.

• **Verifying factors that strengthened low-frequency components.** *(1) Verifying that a deep network strengthened low-frequency components in Remark 1 and Remark 2.* To this end, we constructed a network with 50 convolutional layers. Each convolutional layer applied zero-paddings to avoid changing the size of feature maps, and was followed by an ReLU layer. We conducted this experiment on three datasets, including CIFAR-10 (Krizhevsky et al., 2009), Tiny-ImageNet, and Broden (Bau et al., 2017) datasets, respectively. The exponential increase of $\mathbb{T}^{(uv)(L:1)}$ along with the network depth $L$ in Remark 1 indicates that frequency component $\mathbf{h}^{(uv)}$ of the network output also increases exponentially along with $L$. Therefore, for the frequency component $\mathbf{h}^{(uv)}$ generated

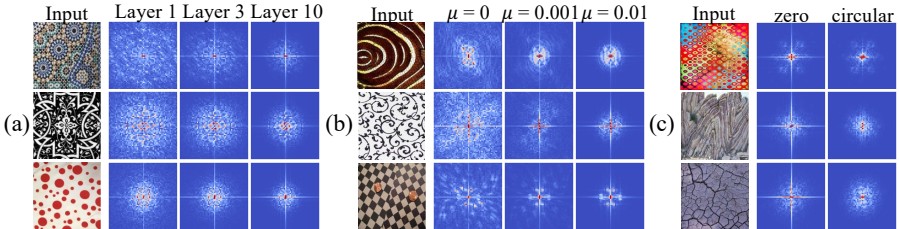

Figure 5: (a) A higher layer of a network usually generated features with more low-frequency components, but with less high-frequency components. (b) A network whose convolutional weights have a mean value significantly biased from 0 usually strengthened low-frequency components, but weakened high-frequency components. (c) A network with zero-padding operations usually strengthened more low-frequency components than a network with circular padding operations. Here, each magnitude map of the feature spectrum was averaged over all channels. For clarity, we moved low frequencies to the center of the spectrum map, and moved high frequencies to corners of the spectrum map. Besides, we only visualized components in the center of the spectrum map with the range of relatively low frequencies $u \in \{u | 0 \leq u < M/8\} \cup \{u | 7M/8 \leq u < M\}; v \in \{v | 0 \leq v < N/8\} \cup \{v | 7N/8 \leq v < N\}$ for clarity.

by each $l$-th layer in a real decoder network, we measured its second-order moment $SOM(\mathbf{h}^{(uv)})$. Figure 4(a) shows that $SOM(\mathbf{h}^{(uv)})$ increased along with the layer number in an exponential manner.

Besides, we visualized feature spectrums of different convolutional layers, which verified the claim in Remark 2 that a deep decoder network strengthens the encoding of low-frequency components of the input sample. Results on the Broden dataset in Figure 5(a) show that magnitudes of low-frequency components increased along with the network layer number. In addition, Appendix C.4 shows results on more datasets, which also yielded similar conclusions.

*(2) Verifying that a larger absolute mean value $\mu_l$ of each $l$-th layer's parameters strengthened low-frequency components in Remark 3.* To this end, we compared feature spectrums of the same network architecture with different mean values $\mu_l$ of parameters. Therefore, we applied the network architecture used in the verification of the effects of the zero-padding, but we changed the kernel size to 9×9. Based on this architecture, we constructed three networks, whose parameters were sampled from Gaussian distributions $\mathcal{N}(\mu = 0, \sigma^2 = 0.01^2)$, $\mathcal{N}(\mu = 0.001, \sigma^2 = 0.01^2)$, and $\mathcal{N}(\mu = 0.01, \sigma^2 = 0.01^2)$, respectively. Results on the Broden dataset in Figure 5(b) show that magnitudes of low-frequency components increased along with the absolute mean value of parameters. In addition, Appendix C.5 shows results on more datasets, which also yielded similar conclusions.

*(3) Verifying that a small kernel size $K$ strengthened low-frequency components in Remark 4.* To this end, we compared feature spectrums of networks with different kernel sizes. Therefore, we constructed three networks with kernel sizes of 1×1, 3×3, and 5×5. Each network contained 5 convolutional layers, each layer contained 16 convolutional kernels, except for the last layer containing 3 kernels. We used the metric $p^{\text{low}} = \sum_{[u,v] \in \Omega^{\text{low}}} \mathbb{E}_c[|H_{uv}^{(c)}|^2] / \sum_{uv} \mathbb{E}_c[|H_{uv}^{(c)}|^2]$ to measure the ratio of low-frequency components to all frequencies, where $\Omega^{\text{low}} = [0 \leq u < M/8, 0 \leq v < N/8] \cup [0 \leq u < M/8, 7N/8 \leq v < N] \cup [7M/8 \leq u < M, 0 \leq v < N/8] \cup [7M/8 \leq u < M, 7N/8 \leq v < N]$. Figure 4(b) show that the network with a small kernel size encoded more low-frequency components.

## 5 CONCLUSION

In this paper, we have reformulate the rule for the forward propagation of a cascaded convolutional decoder network in the frequency domain. Based on such propagation rules, we have discovered and theoretically proven that both the convolution operation and the zero-padding operation strengthen low-frequency components in the decoder. Besides, the upsampling operation repeats the strong magnitude of the fundamental frequency in the input feature to different frequencies of the spectrum of the output feature map. Such properties may hurt the representation capacity of a convolutional decoder network. Experiments have verified our theoretical proofs. Note that our findings can explain general trends of networks with above three operations, but cannot derive a deterministic property of a specific network, and cannot be extended to networks for image classification, because we have not derived the property of the max-pooling operation, which will be derived in the future.

ETHICS STATEMENT STATEMENT

As a fundamental research in machine learning, this paper does not introduce any new ethical or societal concerns. The results in this paper do not include misleading claims; their correctness is theoretically verified. Related work is accurately represented. Though in theory any technique can be misused, it is not likely to happen at the current stage.

REPRODUCIBILITY STATEMENT

This research discovered and theoretically explained two bottlenecks of a cascaded convolutional decoder network in representing feature spectrums. For our theoretical results, formal statements and the complete proofs of all Theorems, Corollaries, and the Assumption in Section 2 and Section 3 are provided in Appendix A. We have discribed additional experimental details in Appendix C, including various model architectures and benchmark datasets, which ensure the reproducibility. We will release all the codes and datasets when this paper is accepted.

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

# A   PROOFS OF OUR THEORETICAL FINDINGS

We first introduce an important equation, which is widely used in the following proofs.

*Lemma 1. Given $N$ complex numbers, $e^{in\theta}$, $n = 0, 1, \ldots, N - 1$, the sum of these $N$ complex numbers is given as follows.*

$$\forall \theta \in \mathbb{R}, \qquad \sum_{n=0}^{N-1} e^{in\theta} = \frac{\sin(\frac{N\theta}{2})}{\sin(\frac{\theta}{2})} e^{i\frac{(N-1)\theta}{2}} \tag{1}$$

*Specifically, when $N\theta = 2k\pi, k \in \mathbb{Z}, -N < k < N$, we have*

$$\forall \theta \in \mathbb{R}, \quad \sum_{n=0}^{N-1} e^{in\theta} = \frac{\sin(\frac{N\theta}{2})}{\sin(\frac{\theta}{2})} e^{i\frac{(N-1)\theta}{2}} = N\delta_\theta; \quad s.t. \ N\theta = 2k\pi, k \in \mathbb{Z}, -N < k < N,$$
$$where \quad \delta_\theta = \begin{cases} 1, & \theta = 0 \\ 0, & otherwise \end{cases} \tag{2}$$

We prove Lemma 1 as follows.

*Proof.* First, let us use the letter $S \in \mathbb{C}$ to denote the term of $\sum_{n=0}^{N-1} e^{in\theta}$.

$$S = \sum_{n=0}^{N-1} e^{in\theta}$$

Therefore, $e^{i\theta}S$ is formulated as follows.

$$e^{i\theta}S = \sum_{n=1}^{N} e^{in\theta} \in \mathbb{C}$$

Then, $S$ can be computed as $S = \frac{e^{i\theta}S - S}{e^{i\theta} - 1}$. Therefore, we have

$$\begin{aligned} S &= \frac{e^{i\theta}S - S}{e^{i\theta} - 1} \\ &= \frac{\sum_{n=1}^{N} e^{in\theta} - \sum_{n=0}^{N-1} e^{in\theta}}{e^{i\theta} - 1} \\ &= \frac{e^{iN\theta} - 1}{e^{i\theta} - 1} \\ &= \frac{e^{i\frac{N\theta}{2}} - e^{-i\frac{N\theta}{2}}}{e^{i\frac{\theta}{2}} - e^{-i\frac{\theta}{2}}} e^{i\frac{(N-1)\theta}{2}} \\ &= \frac{(e^{i\frac{N\theta}{2}} - e^{-i\frac{N\theta}{2}})/2i}{(e^{i\frac{\theta}{2}} - e^{-i\frac{\theta}{2}})/2i} e^{i\frac{(N-1)\theta}{2}} \\ &= \frac{\sin(\frac{N\theta}{2})}{\sin(\frac{\theta}{2})} e^{i\frac{(N-1)\theta}{2}} \end{aligned}$$

Therefore, we prove that $\sum_{n=0}^{N-1} e^{in\theta} = \frac{\sin(\frac{N\theta}{2})}{\sin(\frac{\theta}{2})} e^{i\frac{(N-1)\theta}{2}}$.

Then, we prove the special case that when $N\theta = 2k\pi, k \in \mathbb{Z}, -N < k < N, \sum_{n=0}^{N-1} e^{in\theta} = N\delta_\theta = \begin{cases} N, & \theta = 0 \\ 0, & otherwise \end{cases}$, as follows.

When $\theta = 0$, we have

$$\lim_{\theta \to 0} \sum_{n=0}^{N-1} e^{in\theta} = \lim_{\theta \to 0} \frac{\sin(\frac{N\theta}{2})}{\sin(\frac{\theta}{2})} e^{i\frac{(N-1)\theta}{2}}$$

$$= \lim_{\theta \to 0} \frac{\sin(\frac{N\theta}{2})}{\sin(\frac{\theta}{2})}$$

$$= N$$

When $\theta \neq 0$, and $N\theta = 2k\pi, k \in \mathbb{Z}, -N < k < N$, we have

$$\sum_{n=0}^{N-1} e^{in\theta} = \frac{\sin(\frac{N\theta}{2})}{\sin(\frac{\theta}{2})} e^{i\frac{(N-1)\theta}{2}}$$

$$= \frac{\sin(k\pi)}{\sin(\frac{k\pi}{N})} e^{i\frac{(N-1)k\pi}{N}}$$

$$= 0$$

$\square$

In the following proofs, the following two equations are widely used, which are derived based on Lemma 1.

$$\sum_{m=0}^{M-1} \sum_{n=0}^{N-1} e^{-i(\frac{um}{M} + \frac{vn}{N})2\pi} = \sum_{m=0}^{M-1} e^{im(-\frac{u2\pi}{M})} \sum_{n=0}^{N-1} e^{in(-\frac{v2\pi}{N})}$$

$$= (M\delta_{-\frac{u2\pi}{M}})(N\delta_{-\frac{v2\pi}{N}}) \quad //\text{According to Equation (2)}$$

$$= \begin{cases} MN, & u = v = 0 \\ 0, & \text{otherwise} \end{cases}$$

To simplify the representation, **let $\delta_{uv}$ be the simplification of $\delta_{-\frac{u2\pi}{M}} \delta_{-\frac{v2\pi}{N}}$ in the following proofs.** Therefore, we have

$$\sum_{m=0}^{M-1} \sum_{n=0}^{N-1} e^{-i(\frac{um}{M} + \frac{vn}{N})2\pi} = MN\delta_{uv} = \begin{cases} MN, & u = v = 0 \\ 0, & \text{otherwise} \end{cases} \tag{3}$$

Similarly, we derive the second equation as follows.

$$\sum_{m=0}^{M-1} \sum_{n=0}^{N-1} e^{i(\frac{(u-u')m}{M} + \frac{(v-v')n}{N})2\pi} = \sum_{m=0}^{M-1} e^{im(\frac{(u-u')2\pi}{M})} \sum_{n=0}^{N-1} e^{in(\frac{(v-v')2\pi}{N})}$$

$$= MN\delta_{\frac{(u-u')2\pi}{M}} \delta_{\frac{(v-v')2\pi}{N}} \quad //\text{According to Equation (2)}$$

$$= MN\delta_{u-u'}\delta_{v-v'} \tag{4}$$

$$= \begin{cases} MN, & u' = u; v' = v \\ 0, & \text{otherwise} \end{cases}$$

### A.1 PROOF OF THEOREM 1

In this section, we prove Theorem 1 in Section 2 of the main paper.

*Theorem* 1. *Let $\boldsymbol{H} = [H^{(1)}, H^{(2)}, \ldots, H^{(D)}] \in \mathbb{C}^{D \times M' \times N'}$ denote spectrums of the output feature $\widetilde{\boldsymbol{F}} \in \mathbb{R}^{D \times M' \times N'}$. Then, $\boldsymbol{H}$ can be computed as follows.*

$$\boldsymbol{h}^{(u'v')} = \delta_{u'v'} MN\boldsymbol{b} + \sum_{u=0}^{M-1} \sum_{v=0}^{N-1} \alpha_{u'v'uv} \cdot T^{(uv)} \boldsymbol{g}^{(uv)}, \qquad s.t. \ \delta_{u'v'} = \begin{cases} 1, & u' = 0; v' = 0 \\ 0, & otherwise \end{cases};$$

$$\alpha_{u'v'uv} = \frac{1}{MN} \frac{\sin((M-K)\lambda_{uu'}\pi)}{\sin(\lambda_{uu'}\pi)} \frac{\sin((N-K)\gamma_{vv'}\pi)}{\sin(\gamma_{vv'}\pi)} e^{i((M-K)\lambda_{uu'}+(N-K)\gamma_{vv'})\pi};$$

*where* $\boldsymbol{h}^{(u'v')} = [H^{(1)}_{u'v'}, H^{(2)}_{u'v'}, \ldots, H^{(D)}_{u'v'}]^\top \in \mathbb{C}^D$; $\boldsymbol{b} = [b^{(1)}, b^{(2)}, \ldots, b^{(D)}]^\top \in \mathbb{R}^D$ *denotes the vector of bias terms;* $\alpha_{u'v'uv} \in \mathbb{C}$ *is a coefficient;* $\lambda_{uu'} = \frac{(u-u')M-u(K-1)}{M(M-K+1)}$, $\gamma_{vv'} = \frac{(v-v')N-v(K-1)}{N(N-K+1)}$. $T^{(uv)} \in \mathbb{C}^{D\times C}$ *is a matrix of complex numbers, which is exclusively determined by convolutional kernels* $\boldsymbol{W}^{[ker=1]}, \boldsymbol{W}^{[ker=2]}, \ldots, \boldsymbol{W}^{[ker=D]}$.

$$T^{(uv)}_{dc} = \sum_{t=0}^{K} \sum_{s=0}^{K} W^{[ker=d]}_{cts} e^{i(\frac{ut}{M}+\frac{vs}{N})2\pi}, \qquad d = 1, 2, \ldots, D; \ c = 1, 2, \ldots, C.$$

*Proof.* Given each $c$-th channel of the feature spectrum $G^{(c)}$, the corresponding feature $F^{(c)}$ in the time domain can be computed as follows.

$$F^{(c)}_{mn} = \frac{1}{MN} \sum_{u=0}^{M-1} \sum_{v=0}^{N-1} G^{(c)}_{uv} e^{i(\frac{um}{M}+\frac{vn}{N})2\pi}$$

Then, let us conduct the convlution operation (in Equation (1) in the main paper) on feature $\mathbf{F} = [F^{(1)}, F^{(2)}, \ldots, F^{(C)}]$, in order to obtain the output feature $\widetilde{\mathbf{F}} \in \mathbb{R}^{D\times M'\times N'}$.

$$\forall d = 1, 2, \ldots, D; 0 \le m < M'; 0 \le n < N';$$

$$\begin{aligned}
\tilde{F}^{(d)}_{mn} &= b^{(d)} + \sum_{c=1}^{C} \sum_{t=0}^{K-1} \sum_{s=0}^{K-1} W^{ker=d}_{cts} F^{(c)}_{m+t,n+s} \\
&= b^{(d)} + \sum_{c=1}^{C} \sum_{t=0}^{K-1} \sum_{s=0}^{K-1} W^{ker=d}_{cts} \frac{1}{MN} \sum_{u=0}^{M-1} \sum_{v=0}^{N-1} G^{(c)}_{uv} e^{i(\frac{u(m+t)}{M}+\frac{v(n+s)}{N})2\pi} \\
&= b^{(d)} + \sum_{c=1}^{C} \frac{1}{MN} \sum_{u=0}^{M-1} \sum_{v=0}^{N-1} G^{(c)}_{uv} e^{i(\frac{um}{M}+\frac{vn}{N})2\pi} \sum_{t=0}^{K-1} \sum_{s=0}^{K-1} W^{ker=d}_{cts} e^{i(\frac{ut}{M}+\frac{vs}{N})2\pi} \\
&= b^{(d)} + \sum_{c=1}^{C} \frac{1}{MN} \sum_{u=0}^{M-1} \sum_{v=0}^{N-1} T^{(uv)}_{dc} G^{(c)}_{uv} e^{i(\frac{um}{M}+\frac{vn}{N})2\pi}
\end{aligned}$$

Then, let us conduct the DFT on each channel of $\widetilde{\mathbf{F}}$, in order to obtain feature spectrums $H^{(d)}_{u'v'}$ of $\widetilde{\mathbf{F}}$.

$$\forall d = 1, 2, \ldots, D; \ 0 \le u' < M'; \ 0 \le v' < N';$$

$$\begin{aligned}
H^{(d)}_{u'v'} &= \sum_{m=0}^{M'-1} \sum_{n=0}^{N'-1} \widetilde{F}^{(l,d)}_{mn} e^{-i(\frac{u'm}{M'}+\frac{v'n}{N'})2\pi} \\
&= \sum_{m=0}^{M'-1} \sum_{n=0}^{N'-1} e^{-i(\frac{u'm}{M'}+\frac{v'n}{N'})2\pi} \left(b^{(d)} + \sum_{c=1}^{C} \frac{1}{MN} \sum_{u=0}^{M-1} \sum_{v=0}^{N-1} T^{(uv)}_{dc} G^{(c)}_{uv} e^{i(\frac{um}{M}+\frac{vn}{N})2\pi}\right) \quad //\text{Equation (3)} \\
&= M'N'b^{(d)}\delta_{u'v'} + \sum_{c=1}^{C} \sum_{u=0}^{M-1} \sum_{v=0}^{N-1} T^{(uv)}_{dc} G^{(c)}_{uv} \frac{1}{MN} \sum_{m=0}^{M'-1} \sum_{n=0}^{N'-1} e^{i((\frac{u}{M}-\frac{u'}{M'})m+(\frac{v}{N}-\frac{v'}{N'})n)2\pi}
\end{aligned}$$

$$// \text{ Let } \alpha_{u'v'uv} = \frac{1}{MN} \sum_{m=0}^{M'-1} \sum_{n=0}^{N'-1} e^{i((\frac{u}{M}-\frac{u'}{M'})m+(\frac{v}{N}-\frac{v'}{N'})n)2\pi}$$

$$= M'N'b^{(d)}\delta_{u'v'} + \sum_{u=0}^{M-1} \sum_{v=0}^{N-1} \alpha_{u'v'uv} \sum_{c=1}^{C} T^{(uv)}_{dc} G^{(c)}_{uv}$$

When the convlution operation does not apply paddings, and its stride size is 1, $M' = M - K + 1$, $N' = N - K + 1$. In this way, $\alpha_{u'v'uv}$ can be rewritten as follows.

$$
\begin{aligned}
\alpha_{u'v'uv} &= \frac{1}{MN} \sum_{m=0}^{M'-1} \sum_{n=0}^{N'-1} e^{i((\frac{u}{M} - \frac{u'}{M'})m + (\frac{v}{N} - \frac{v'}{N'})n)2\pi} \\
&//M' = M - K + 1, N' = N - K + 1 \\
&= \frac{1}{MN} \sum_{m=0}^{M-K} \sum_{n=0}^{N-K} e^{i((\frac{u}{M} - \frac{u'}{M-K+1})m + (\frac{v}{N} - \frac{v'}{N-K+1})n)2\pi} \\
&= \frac{1}{MN} \sum_{m=0}^{M-K} e^{i(\frac{u}{M} - \frac{u'}{M-K+1})2\pi m} \sum_{n=0}^{N-K} e^{i(\frac{v}{N} - \frac{v'}{N-K+1})2\pi n} \\
&//\text{According to Equation (1)} \\
&= \frac{1}{MN} \frac{\sin((M-K)\lambda_{uu'}\pi)}{\sin(\lambda_{uu'}\pi)} \frac{\sin((N-K)\gamma_{vv'}\pi)}{\sin(\gamma_{vv'}\pi)} e^{i((M-K)\lambda_{uu'} + (N-K)\gamma_{vv'})\pi}
\end{aligned}
$$
(5)

where $\lambda_{uu'} = \frac{(u-u')M - u(K-1)}{M(M-K+1)}$, $\gamma_{vv'} = \frac{(v-v')N - v(K-1)}{N(N-K+1)}$.

Therefore, we prove that the vector $\mathbf{h}^{(u'v')} = [H_{u'v'}^{(1)}, H_{u'v'}^{(2)}, \ldots, H_{u'v'}^{(D)}]^\top \in \mathbb{C}^D$ can be computed as follows.

$$
\forall d = 1, 2, \ldots, D; \quad \mathbf{h}^{(u'v')} = \delta_{u'v'} M'N'\mathbf{b} + \sum_{u=0}^{M-1} \sum_{v=0}^{N-1} \alpha_{u'v'uv} T^{(uv)} \mathbf{g}^{(uv)}
$$

$\square$

## A.2 PROOF OF THEOREM 2

In this section, we prove Theorem 2 in Section 2 of the main paper.

*Theorem 2. Based on Assumption 1, the layerwise dynamics of feature spectrums in the frequency domain can be simplified as follows.*

$$
\boldsymbol{h}^{(uv)} = T^{(uv)} \boldsymbol{g}^{(uv)} + \delta_{uv} MN\boldsymbol{b}
$$
(6)

*Proof.* Based on Assumption 1, the convolution operation does not change the size of the feature map, *i.e.*, $M' = M$, $N' = N$. In this case, $\alpha_{u'v'uv}$ can be computed as follows.

$$
\begin{aligned}
\alpha_{u'v'uv} &= \frac{1}{MN} \sum_{m=0}^{M'-1} \sum_{n=0}^{N'-1} e^{i((\frac{u}{M} - \frac{u'}{M'})m + (\frac{v}{N} - \frac{v'}{N'})n)2\pi} \\
&= \frac{1}{MN} \sum_{m=0}^{M-1} \sum_{n=0}^{N-1} e^{i(\frac{(u-u')m}{M} + \frac{(v-v')n}{N})2\pi} \quad //M' = M, N' = N \\
&= \frac{1}{MN} \sum_{m=0}^{M-1} e^{i(\frac{(u-u')2\pi}{M})m} \sum_{n=0}^{N-1} e^{i(\frac{(v-v')2\pi}{N})n} \quad //\text{According to Equation (4)} \\
&= \delta_{u-u'} \delta_{v-v'}
\end{aligned}
$$
(7)

where $\delta_{u-u'} = \begin{cases} 1, u' = u \\ 0, \text{otherwise} \end{cases}$; $\delta_{v-v'} = \begin{cases} 1, v' = v \\ 0, \text{otherwise} \end{cases}$.

Therefore, $\mathbf{h}^{(u'v')}$ can be computed as follows.

$$
\begin{aligned}
\mathbf{h}^{(u'v')} &= \sum_{u=0}^{M'-1} \sum_{v=0}^{N'-1} \alpha_{u'v'uv} T^{(u'v')} \mathbf{g}^{(u'v')} + \delta_{u'v'} M'N'\mathbf{b} \\
&= \sum_{u=0}^{M-1} \sum_{v=0}^{N-1} \delta_{u-u'} \delta_{v-v'} T^{(u'v')} \mathbf{g}^{(u'v')} + \delta_{u'v'} MN\mathbf{b} \\
&= T^{(u'v')} \mathbf{g}^{(u'v')} + MN\mathbf{b}\delta_{u'v'}
\end{aligned}
$$

Then, we prove that $\mathbf{h}^{(uv)} = T^{(uv)}\mathbf{g}^{(uv)} + MN\mathbf{b}\delta_{uv}$. $\qquad\qquad\qquad\qquad\qquad\square$

## A.3 Proof of Corollary 1

In this section, we prove Corollary 1 in Section 2 of the main paper.

*Corollary 1. Let $\boldsymbol{G} = [G^{(1)}, G^{(2)}, \ldots, G^{(C_0)}] \in \mathbb{C}^{C_0 \times M \times N}$ denote frequency spectrums of the $C_0$ channels of $x$. Then, based on Assumption 1, spectrums of the generated image $\widehat{x}$, i.e., $\boldsymbol{H} = [H^{(1)}, H^{(2)}, \ldots, H^{(C_L)}] \in \mathbb{C}^{C_L \times M \times N}$, can be computed as follows.*

$$\boldsymbol{h}^{(uv)} = \mathbb{T}^{(uv)(L:1)}\boldsymbol{g}^{(uv)} + \delta_{uv}\boldsymbol{\beta} \qquad (8)$$

*where $\mathbb{T}^{(uv)(L:1)} = T^{(L,uv)} \cdots T^{(2,uv)}T^{(1,uv)} \in \mathbb{C}^{C_L \times C_0}$; $\boldsymbol{g}^{(uv)} = [G_{uv}^{(1)}, G_{uv}^{(2)}, \ldots, G_{uv}^{(C_0)}]^\top \in \mathbb{C}^{C_0}$ and $\boldsymbol{h}^{(uv)} = [H_{uv}^{(1)}, H_{uv}^{(2)}, \ldots, H_{uv}^{(C_L)}]^\top \in \mathbb{C}^{C_L}$ denote vectors at the frequency $[u, v]$ in tensors $\boldsymbol{G}$ and $\boldsymbol{H}$, respectively. $\boldsymbol{\beta} = MN\big(\boldsymbol{b}^{(L)} + \sum_{j=2}^{L} \mathbb{T}^{(00)(L:j)}\boldsymbol{b}^{(j-1)}\big) \in \mathbb{C}^{C_L}$; $\boldsymbol{b}^{(l)} = [b^{(l,1)}, b^{(l,2)}, \ldots, b^{(l,C_l)}]^\top \in \mathbb{R}^{C_l}$ denotes the vector of bias terms of $C_l$ convolutional kernels in the $l$-th layer.*

*Proof. Let $\boldsymbol{G}^{(l)} = [G^{(l,1)}, G^{(l,2)}, \cdots, G^{(l,C_l)}] \in \mathbb{C}^{C_l \times M \times N}$ denote feature spectrums of the $l$-th layer. Let $\boldsymbol{g}^{(l,uv)} = [G_{uv}^{(l,1)}, G_{uv}^{(l,2)}, \cdots, G_{uv}^{(l,C_l)}]^\top \in \mathbb{C}^{C_l}$ denote the frequency component at the frequency $[u, v]$. When $l = 0$, $\mathbf{g}^{(0,uv)}$ denotes the frequency component of the input sample. When $l = L$, $\mathbf{g}^{(L,uv)}$ denotes the frequency component of the network output.*

Based on Theorem 2, $\mathbf{g}^{(l,uv)}$ can be computed as follows.

$$\forall l = 1, 2, \ldots, L, \quad \mathbf{g}^{(l,uv)} = T^{(l,uv)}\mathbf{g}^{(l-1,uv)} + \delta_{uv}MN\mathbf{b}^{(l)}$$

Then, the frequency component $\mathbf{g}^{(L,uv)}$ of the network output can be computed as follows.

$$\begin{aligned}
\mathbf{g}^{(L,uv)} &= T^{(L,uv)}\mathbf{g}^{(L-1,uv)} + \delta_{uv}MN\mathbf{b}^{(L)} \\
&= T^{(L,uv)}(T^{(L-1,uv)}\mathbf{g}^{(L-2,uv)} + \delta_{uv}MN\mathbf{b}^{(L-1)}) + \delta_{uv}MN\mathbf{b}^{(L)} \\
&= T^{(L,uv)}T^{(L-1,uv)}\mathbf{g}^{(L-2,uv)} + T^{(L,uv)}\delta_{uv}MN\mathbf{b}^{(L-1)} + \delta_{uv}MN\mathbf{b}^{(L)} \\
&= \cdots \\
&= T_{dc}^{(l,uv)}\cdots T^{(1,uv)}\mathbf{g}^{(0,uv)} + MNT_{dc}^{(l,uv)}\cdots T^{(2,uv)}\mathbf{b}^{(1)}\delta_{uv} + \cdots + MN\mathbf{b}^{(L)}\delta_{uv} \\
&= T_{dc}^{(l,uv)}\cdots T^{(1,uv)}\mathbf{g}^{(0,uv)} + \delta_{uv}MN(T_{dc}^{(l,uv)}\cdots T^{(2,uv)}\mathbf{b}^{(1)} + \cdots + MN\mathbf{b}^{(L)})
\end{aligned}$$

Let $\mathbb{T}^{(uv)(L:1)} = T_{dc}^{(l,uv)}\cdots T^{(2,uv)}T^{(1,uv)}$ and $\boldsymbol{\beta} = MN\big(\mathbf{b}^{(L)} + \sum_{j=2}^{L} \mathbb{T}^{(00)(L:j)}\mathbf{b}^{(j-1)}\big)$. Let $\mathbf{h}^{(uv)} = \mathbf{g}^{(L,uv)}$ denote the frequency component of the network output, and let $\mathbf{g}^{(uv)} = \mathbf{g}^{(0,uv)}$ denote the frequency component of the input sample. Then, we prove that $\mathbf{h}^{(uv)}$ can be computed as follows.

$$\mathbf{h}^{(uv)} = \mathbb{T}^{(uv)(L:1)}\mathbf{g}^{(uv)} + \delta_{uv}\boldsymbol{\beta}$$

$\qquad\qquad\qquad\qquad\qquad\qquad\qquad\qquad\qquad\qquad\qquad\qquad\qquad\qquad\square$

## A.4 Proof of Corollary 2

In this section, we prove Corollary 2 in Section 2 of the main paper.

*Corollary 2. Based on Assumption 1, the change of each frequency components $T^{(l,uv)}$ during the learning process is reformulated as follows.*

$$\Delta T^{(l,uv)} = -\eta MN \sum_{u'=0}^{M-1}\sum_{v'=0}^{N-1} \chi_{u'v'uv} \left(\overline{\mathbb{T}}^{(u'v')(l-1:1)}\overline{\boldsymbol{g}}^{(u'v')} + \delta_{u'v'}\overline{\boldsymbol{\beta}}'\right) \frac{\partial Loss}{\partial(\overline{\boldsymbol{h}}^{(u'v')})^\top}\overline{\mathbb{T}}^{(u'v')(L:l+1)}; \quad (9)$$

$$s.t. \quad \chi_{u'v'uv} = \frac{1}{MN}\frac{\sin(K(u-u')\pi/M)}{\sin((u-u')\pi/M)}\frac{\sin(K(v-v')\pi/N)}{\sin((v-v')\pi/N)}e^{i(\frac{(K-1)(u-u')}{M} + \frac{(K-1)(v-v')}{N})\pi} \quad (10)$$

*where $\eta$ is the learning rate; $\chi_{u'v'uv} \in \mathbb{C}$ is a coefficient; $\mathbb{T}^{(u'v')(l-1:1)} = T^{(l-1,u'v')}\cdots T^{(2,u'v')}T^{(1,u'v')} \in \mathbb{C}^{C_{l-1}\times C_0}$; $\mathbb{T}^{(u'v')(L:l+1)} = T^{(L,u'v')}\cdots T^{(l+1,u'v')} \in \mathbb{C}^{C_L \times C_l}$; $\boldsymbol{\beta}' = MN\big(\boldsymbol{b}^{(l-1)} + \sum_{j=2}^{l-1}\mathbb{T}^{(00)(l-1:j)}\boldsymbol{b}^{(j-1)}\big) \in \mathbb{C}^{C_{l-1}}$; $\overline{\mathbb{T}}^{(uv)(l-1:1)}$ is the conjugate of $\mathbb{T}^{(uv)(l-1:1)}$.*

*Proof.* **First, we focus on a single convolutional layer.**

According to the DFT and the inverse DFT, we can obtain the mathematical relationship between $G_{uv}^{(l,c)}$ and $F_{mn}^{(l,c)}$, and the mathematical relationship between $T_{dc}^{(l,uv)}$ and $W_{cts}^{(l)[\text{ker}=d]}$, as follows.

$$
\begin{cases}
G_{uv}^{(l,c)} = \sum_{m=0}^{M-1} \sum_{n=0}^{N-1} F_{mn}^{(l,c)} e^{-i(\frac{um}{M} + \frac{vn}{N})2\pi} \\
F_{mn}^{(l,c)} = \dfrac{1}{MN} \sum_{u=0}^{M-1} \sum_{v=0}^{N-1} G_{uv}^{(l,c)} e^{i(\frac{um}{M} + \frac{vn}{N})2\pi}
\end{cases}
\begin{cases}
T_{dc}^{(l,uv)} = \sum_{t=0}^{K-1} \sum_{s=0}^{K-1} W_{cts}^{(l)[\text{ker}=d]} e^{i(\frac{ut}{M} + \frac{vs}{N})2\pi} \\
W_{cts}^{(l)[\text{ker}=d]} = \dfrac{1}{MN} \sum_{u=0}^{M-1} \sum_{v=0}^{N-1} T_{dc}^{(l,uv)} e^{-i(\frac{ut}{M} + \frac{vs}{N})2\pi}
\end{cases}
\tag{11}
$$

Based on Equation (11) and the derivation rule for complex numbers (Kreutz-Delgado, 2009), we can obtain the mathematical relationship between $\frac{\partial Loss}{\partial \overline{G}_{uv}^{(l,c)}}$ and $\frac{\partial Loss}{\partial \overline{F}_{mn}^{(l,c)}}$, and the mathematical relationship between $\frac{\partial Loss}{\partial \overline{T}_{dc}^{(l,uv)}}$ and $\frac{\partial Loss}{\partial \overline{W}_{cts}^{(l)[\text{ker}=d]}}$, as follows. Note that when we use gradient descent to optimize a real-valued loss function *Loss* with complex variables, people usually treat the real and imaginary values, $a \in \mathbb{C}$ and $b \in \mathbb{C}$, of a complex variable ($z = a + bi$) as two separate real-valued variables, and separately update these two real-valued variables. In this way, the exact optimization step of $z$ computed based on such a technology is equivalent to $\frac{\partial Loss}{\partial \overline{z}}$. Since $F_{mn}^{(l,c)}$ and $W_{cts}^{(l)[\text{ker}=d]}$ are real numbers, $\frac{\partial Loss}{\partial \overline{F}_{mn}^{(l,c)}} = \frac{\partial Loss}{\partial F_{mn}^{(l,c)}}$ and $\frac{\partial Loss}{\partial \overline{W}_{cts}^{(l)[\text{ker}=d]}} = \frac{\partial Loss}{\partial W_{cts}^{(l)[\text{ker}=d]}}$.

$$
\begin{cases}
\dfrac{\partial Loss}{\partial \overline{G}_{uv}^{(l,c)}} = \dfrac{1}{MN} \sum_{m=0}^{M-1} \sum_{n=0}^{N-1} \dfrac{\partial Loss}{\partial \overline{F}_{mn}^{(l,c)}} e^{-i(\frac{um}{M} + \frac{vn}{N})2\pi} \\
\dfrac{\partial Loss}{\partial \overline{F}_{mn}^{(l,c)}} = \sum_{u=0}^{M-1} \sum_{v=0}^{N-1} \dfrac{\partial Loss}{\partial \overline{G}_{uv}^{(l,c)}} e^{i(\frac{um}{M} + \frac{vn}{N})2\pi}
\end{cases}
\begin{cases}
\dfrac{\partial Loss}{\partial \overline{T}_{dc}^{(l,uv)}} = \dfrac{1}{MN} \sum_{t=0}^{K-1} \sum_{s=0}^{K-1} \dfrac{\partial Loss}{\partial \overline{W}_{cts}^{(l)[\text{ker}=d]}} e^{i(\frac{ut}{M} + \frac{vs}{N})2\pi} \\
\dfrac{\partial Loss}{\partial \overline{W}_{cts}^{(l)[\text{ker}=d]}} = \sum_{u=0}^{M-1} \sum_{v=0}^{N-1} \dfrac{\partial Loss}{\partial \overline{T}_{dc}^{(l,uv)}} e^{-i(\frac{ut}{M} + \frac{vs}{N})2\pi}
\end{cases}
\tag{12}
$$

Let us conduct the convolution operation (based on Assumption 1) on the feature map $\mathbf{F}^{(l-1)} = [F^{(l-1,1)}, F^{(l-1,2)}, \ldots, F^{(l-1,C)}] \in \mathbb{R}^{C \times M \times N}$, and obtain the output feature map $\mathbf{F}^{(l)} = [F^{(l,1)}, F^{(l,2)}, \ldots, F^{(l,D)}] \in \mathbb{R}^{D \times M \times N}$ of the l-th layer as follows.

$$
F_{mn}^{(l,d)} = b^{(d)} + \sum_{c=1}^{C} \sum_{t=0}^{K-1} \sum_{s=0}^{K-1} W_{cts}^{(l)[\text{ker}=d]} F_{m+t,n+s}^{(l-1,c)}
\tag{13}
$$

Based on Equation (11) and Equation (12), and the derivation rule for complex numbers (Kreutz-Delgado, 2009), the exact optimization step of $T_{dc}^{(l,uv)}$ in real implementations can be computed as

follows.

$$\frac{\partial Loss}{\partial \overline{T}_{dc}^{(l,uv)}}$$

$$= \frac{1}{MN} \sum_{t=0}^{K-1} \sum_{s=0}^{K-1} \frac{\partial Loss}{\partial \overline{W}_{cts}^{(l)[ker=d]}} e^{i(\frac{ut}{M} + \frac{vs}{N})2\pi} \quad //\text{Equation (12)}$$

$$= \frac{1}{MN} \sum_{t=0}^{K-1} \sum_{s=0}^{K-1} \left( \sum_{m=0}^{M-1} \sum_{n=0}^{N-1} \frac{\partial Loss}{\partial \overline{F}_{mn}^{(l,d)}} \cdot \overline{F}_{m+t,n+s}^{(l-1,c)} \right) e^{i(\frac{ut}{M} + \frac{vs}{N})2\pi} \quad //\text{Equation (13)}$$

$$//\text{Equation (11)}$$

$$= \frac{1}{MN} \sum_{t=0}^{K-1} \sum_{s=0}^{K-1} \left( \sum_{m=0}^{M-1} \sum_{n=0}^{N-1} \frac{\partial Loss}{\partial \overline{F}_{mn}^{(l,d)}} \cdot \frac{1}{MN} \sum_{u'=0}^{M-1} \sum_{v'=0}^{N-1} \overline{G}_{u'v'}^{(l-1,c)} e^{-i(\frac{u'(m+t)}{M} + \frac{v'(n+s)}{N})2\pi} \right) e^{i(\frac{ut}{M} + \frac{vs}{N})2\pi}$$

$$= \frac{1}{MN} \sum_{t=0}^{K-1} \sum_{s=0}^{K-1} \left( \sum_{u'=0}^{M-1} \sum_{v'=0}^{N-1} \overline{G}_{u'v'}^{(l-1,c)} e^{-i(\frac{u't}{M} + \frac{v's}{N})2\pi} \cdot \frac{1}{MN} \sum_{m=0}^{M-1} \sum_{n=0}^{N-1} \frac{\partial Loss}{\partial \overline{F}_{mn}^{(l,d)}} e^{-i(\frac{u'm}{M} + \frac{v'n}{N})2\pi} \right) e^{i(\frac{ut}{M} + \frac{vs}{N})2\pi}$$

$$= \frac{1}{MN} \sum_{t=0}^{K-1} \sum_{s=0}^{K-1} \left( \sum_{u'=0}^{M-1} \sum_{v'=0}^{N-1} \overline{G}_{u'v'}^{(l-1,c)} \frac{\partial Loss}{\partial \overline{G}_{u'v'}^{(l,d)}} e^{-i(\frac{u't}{M} + \frac{vs}{N})2\pi} \right) e^{i(\frac{ut}{M} + \frac{vs}{N})2\pi} \quad //\text{Equation (12)}$$

$$= \frac{1}{MN} \sum_{t=0}^{K-1} \sum_{s=0}^{K-1} \sum_{u'=0}^{M-1} \sum_{v'=0}^{N-1} \overline{G}_{u'v'}^{(l-1,c)} \frac{\partial Loss}{\partial \overline{G}_{u'v'}^{(l,d)}} e^{i(\frac{(u-u')t}{M} + \frac{(v-v')s}{N})2\pi}$$

$$= \sum_{u'=0}^{M-1} \sum_{v'=0}^{N-1} \overline{G}_{u'v'}^{(l-1,c)} \frac{\partial Loss}{\partial \overline{G}_{u'v'}^{(l,d)}} \cdot \frac{1}{MN} \sum_{t=0}^{K-1} \sum_{s=0}^{K-1} e^{i(\frac{(u-u')t}{M} + \frac{(v-v')s}{N})2\pi}$$

$$// \text{ Let } \chi_{u'v'uv} = \frac{1}{MN} \sum_{t=0}^{K-1} \sum_{s=0}^{K-1} e^{i(\frac{(u-u')t}{M} + \frac{(v-v')s}{N})2\pi}$$

$$= \sum_{u'=0}^{M-1} \sum_{v'=0}^{N-1} \chi_{u'v'uv} \overline{G}_{u'v'}^{(l-1,c)} \frac{\partial Loss}{\partial \overline{G}_{u'v'}^{(l,d)}}$$

where $\chi_{u'v'uv}$ can be rewritten as follows.

$$\chi_{u'v'uv} = \frac{1}{MN} \sum_{t=0}^{K-1} \sum_{s=0}^{K-1} e^{i(\frac{(u-u')t}{M} + \frac{(v-v')s}{N})2\pi}$$

$$= \frac{1}{MN} \sum_{t=0}^{K-1} e^{i\frac{(u-u')2\pi}{M}t} \sum_{s=0}^{K-1} e^{i\frac{(v-v')2\pi}{N}s}$$

$$= \frac{1}{MN} \frac{\sin(\frac{K(u-u')\pi}{M})}{\sin(\frac{(u-u')\pi}{M})} \frac{\sin(\frac{K(v-v')\pi}{N})}{\sin(\frac{(v-v')\pi}{N})} \cdot e^{i(\frac{(K-1)(u-u')}{M} + \frac{(K-1)(v-v')}{N})\pi} \quad //\text{According to Equation1}$$

Similarly, we computed the gradient of the loss function *w.r.t.* the spectrum map $\overline{G}^{(l-1,c)}$ as follows.

$$
\frac{\partial Loss}{\partial \overline{G}_{u'v'}^{(l-1,c)}}
$$

$$
= \frac{1}{MN} \sum_{m=0}^{M-1} \sum_{n=0}^{N-1} \frac{\partial Loss}{\partial \overline{F}_{mn}^{(l-1,c)}} e^{-i(\frac{u'm}{M} + \frac{v'n}{N})2\pi} \quad //\text{Equation (12)}
$$

$$
= \frac{1}{MN} \sum_{m=0}^{M-1} \sum_{n=0}^{N-1} \left( \sum_{t=0}^{K-1} \sum_{s=0}^{K-1} \overline{W}_{cts}^{(l)[\text{ker}=d]} \cdot \frac{\partial Loss}{\partial \overline{F}_{m-t,n-s}^{(l,d)}} \right) e^{-i(\frac{u'm}{M} + \frac{v'n}{N})2\pi} \quad //\text{Equation (13)}
$$

$$
//\text{Equation (12)}
$$

$$
= \frac{1}{MN} \sum_{m=0}^{M-1} \sum_{n=0}^{N-1} \left( \sum_{t=0}^{K-1} \sum_{s=0}^{K-1} \overline{W}_{cts}^{(l)[\text{ker}=d]} \cdot \sum_{u=0}^{M-1} \sum_{v=0}^{N-1} \frac{\partial Loss}{\partial \overline{G}_{uv}^{(l,d)}} e^{i(\frac{u(m-t)}{M} + \frac{v(n-s)}{N})2\pi} \right) e^{-i(\frac{u'm}{M} + \frac{v'n}{N})2\pi}
$$

$$
= \frac{1}{MN} \sum_{m=0}^{M-1} \sum_{n=0}^{N-1} \left( \sum_{u=0}^{M-1} \sum_{v=0}^{N-1} \frac{\partial Loss}{\partial \overline{G}_{uv}^{(l,d)}} e^{i(\frac{um}{M} + \frac{vn}{N})2\pi} \cdot \sum_{t=0}^{K-1} \sum_{s=0}^{K-1} \overline{W}_{cts}^{(l)[\text{ker}=d]} e^{-i(\frac{ut}{M} + \frac{vs}{N})2\pi} \right) e^{-i(\frac{u'm}{M} + \frac{v'n}{N})2\pi}
$$

$$
= \frac{1}{MN} \sum_{m=0}^{M-1} \sum_{n=0}^{N-1} \left( \sum_{u=0}^{M-1} \sum_{v=0}^{N-1} \frac{\partial Loss}{\partial \overline{G}_{uv}^{(l,d)}} \overline{T}_{dc}^{(l,uv)} e^{i(\frac{um}{M} + \frac{vn}{N})2\pi} \right) e^{-i(\frac{u'm}{M} + \frac{v'n}{N})2\pi} \quad //\text{Equation (11)}
$$

$$
= \sum_{u=0}^{M-1} \sum_{v=0}^{N-1} \frac{\partial Loss}{\partial \overline{G}_{uv}^{(l,d)}} \overline{T}_{dc}^{(l,uv)} \cdot \frac{1}{MN} \sum_{m=0}^{M-1} \sum_{n=0}^{N-1} e^{i(\frac{(u-u')m}{M} + \frac{(v-v')n}{N})2\pi}
$$

$$
= \sum_{u=0}^{M-1} \sum_{v=0}^{N-1} \frac{\partial Loss}{\partial \overline{G}_{uv}^{(l,d)}} \overline{T}_{dc}^{(l,uv)} \cdot \delta_{u-u'} \delta_{v-v'} \quad //\text{Equation (4)}
$$

$$
= \frac{\partial Loss}{\partial \overline{G}_{u'v'}^{(l,d)}} \overline{T}_{dc}^{(l,u'v')}
$$

Based on the derived $\frac{\partial Loss}{\partial \overline{T}_{dc}^{(l,uv)}} \in \mathbb{C}$ and $\frac{\partial Loss}{\partial \overline{G}_{u'v'}^{(l-1,c)}} \in \mathbb{C}$, we can further write gradients $\frac{\partial Loss}{\partial (\overline{T}^{(l,uv)})^{\top}} \in \mathbb{C}^{D \times C}$ and $\frac{\partial Loss}{\partial (\overline{\mathbf{g}}^{(l-1,u'v')})^{\top}} \in \mathbb{C}^{C}$ as follows.

$$
\frac{\partial Loss}{\partial (\overline{T}^{(l,uv)})^{\top}} = \sum_{u'=0}^{M-1} \sum_{v'=0}^{N-1} \chi_{u'v'uv} \overline{\mathbf{g}}^{(l-1,u'v')} \frac{\partial Loss}{\partial (\overline{\mathbf{g}}^{(l,u'v')})^{\top}} \tag{14}
$$

$$
\frac{\partial Loss}{\partial (\overline{\mathbf{g}}^{(l-1,u'v')})^{\top}} = \frac{\partial Loss}{\partial (\overline{\mathbf{g}}^{(l,u'v')})^{\top}} \overline{T}^{(l,u'v')} \tag{15}
$$

Furthermore, **we extend the above proof of a single convolutional layer to a network with $L$ cascaded convolutional layers.** Let $\mathbf{g}^{(l,u'v')}$ denote the frequency component at the frequency $[u', v']$ of the $l$-th layer's output feature, and let $T^{(l,uv)}$ the matrix computed by the $l$-th layer's convolutional weights. Then, according to Equation (15), the gradient *w.r.t.* $\overline{\mathbf{g}}^{(l,u'v')}$ can be computed as follows.

$$
\frac{\partial Loss}{\partial (\overline{\mathbf{g}}^{(l,u'v')})^{T}} = \frac{\partial Loss}{\partial (\overline{\mathbf{g}}^{(L,u'v')})^{T}} \overline{T}^{(L,u'v')} \cdots \overline{T}^{(l+1,u'v')}
$$

$$
= \frac{\partial Loss}{\partial (\overline{\mathbf{g}}^{(L,u'v')})^{T}} \overline{\mathbb{T}}^{(u'v')(L:l+1)} \tag{16}
$$

According to Equation (14), the gradient *w.r.t.* $\overline{T}^{(l,uv)}$ can be computed as follows.

$$\frac{\partial Loss}{\partial (\overline{T}^{(l,uv)})^{\top}} = \sum_{u'=0}^{M-1} \sum_{v'=0}^{N-1} \chi_{u'v'uv} \overline{\mathbf{g}}^{(l-1,u'v')} \frac{\partial Loss}{\partial (\overline{\mathbf{g}}^{(l,u'v')})^{\top}}$$

//Corollary 1

$$= \sum_{u'=0}^{M-1} \sum_{v'=0}^{N-1} \chi_{u'v'uv} (\overline{\mathbb{T}}^{(u'v')(l-1:1)} \overline{\mathbf{g}}^{(0,u'v')} + \overline{\boldsymbol{\beta}}' \delta_{u'v'}) \frac{\partial Loss}{\partial (\overline{\mathbf{g}}^{(L,u'v')})^{\top}} \overline{\mathbb{T}}^{(u'v')(L:l+1)} \quad (17)$$

// Let $\mathbf{g}^{(uv)} = \mathbf{g}^{(0,uv)}; \mathbf{h}^{(uv)} = \mathbf{g}^{(L,uv)}$

$$= \sum_{u'=0}^{M-1} \sum_{v'=0}^{N-1} \chi_{u'v'uv} (\overline{\mathbb{T}}^{(u'v')(l-1:1)} \overline{\mathbf{g}}^{(u'v')} + \overline{\boldsymbol{\beta}}' \delta_{u'v'}) \frac{\partial Loss}{\partial (\overline{\mathbf{h}}^{(u'v')})^{\top}} \overline{\mathbb{T}}^{(u'v')(L:l+1)}$$

Let us use the gradient descent algorithm to update the convlutional weight $W_c^{(l)[\text{ker}=d]}|_n$ of the $n$-th epoch, the updated frequency spectrum $W_c^{(l)[\text{ker}=d]}|_{n+1}$ can be computed as follows.

$$\forall t, s, \quad W_{cts}^{(l)[\text{ker}=d]}|_{n+1} = W_{cts}^{(l)[\text{ker}=d]}|_n - \eta \cdot \frac{\partial Loss}{\partial \overline{W}_{cts}^{(l)[\text{ker}=d]}}$$

where $\eta$ is the learning rate. Then, the updated frequency spectrum $T^{(l,uv)}|_{n+1}$ computed based on Equation (12) is given as follows.

$$\Delta T_{dc}^{(l,uv)} = T_{dc}^{(l,uv)}|_{n+1} - T_{dc}^{(l,uv)}|_n$$

$$= \sum_{t=0}^{K-1} \sum_{s=0}^{K-1} W_{cts}^{(l)[\text{ker}=d]}|_{n+1} e^{i(\frac{ut}{M} + \frac{vs}{N})2\pi} - T_{dc}^{(l,uv)}|_n \quad //\text{Equation (11)}$$

$$= \sum_{t=0}^{K-1} \sum_{s=0}^{K-1} (W_{cts}^{(l)[\text{ker}=d]}|_n - \eta \cdot \frac{\partial Loss}{\partial \overline{W}_{cts}^{(l)[\text{ker}=d]}}) e^{i(\frac{ut}{M} + \frac{vs}{N})2\pi} - T_{dc}^{(l,uv)}|_n$$

$$= (\sum_{t=0}^{K-1} \sum_{s=0}^{K-1} W_{cts}^{(l)[\text{ker}=d]}|_n e^{i(\frac{ut}{M} + \frac{vs}{N})2\pi} - T_{dc}^{(l,uv)}|_n) - \eta \sum_{t=0}^{K-1} \sum_{s=0}^{K-1} \frac{\partial Loss}{\partial \overline{W}_{cts}^{(l)[\text{ker}=d]}} e^{i(\frac{ut}{M} + \frac{vs}{N})2\pi}$$

$$= -\eta \sum_{t=0}^{K-1} \sum_{s=0}^{K-1} \frac{\partial Loss}{\partial \overline{W}_{cts}^{(l)[\text{ker}=d]}} e^{i(\frac{ut}{M} + \frac{vs}{N})2\pi} \quad //\text{Equation (11)}$$

$$= -\eta MN \frac{\partial Loss}{\partial \overline{T}_{dc}^{(l,uv)}} \quad //\text{Equation (12)}$$

Therefore, we prove that any step on $W_{cts}^{(l)[ker=d]}$ equals to $MN$ step on $T_{dc}^{(uv)}$. In this way, pull Equation (17) in the change of $T^{(l,uv)}$ can be computed as follows.

$$\Delta T^{(l,uv)} = -\eta MN \sum_{u'=0}^{M-1} \sum_{v'=0}^{N-1} \chi_{u'v'uv} \left( \overline{\mathbb{T}}^{(u'v')(l-1:1)} \overline{\mathbf{g}}^{(u'v')} + \delta_{u'v'} \overline{\boldsymbol{\beta}}' \right) \frac{\partial Loss}{\partial (\overline{\mathbf{h}}^{(u'v')})^{\top}} \overline{\mathbb{T}}^{(u'v')(L:l+1)} \quad (18)$$

$\square$

## A.5 Proofs of Assumption 2 and Theorem 3

In this section, we prove Assumption 2 and Theorem 3 in the main paper.

*Assumption* 2. *We assume that all elements in $T^{(l,uv)}$ are irrelevant to each other, and $\forall l \neq l'$, elements in $T^{(l,uv)}$ and $T^{(l',uv)}$ are irrelevant to each other in early epochs.*

$$\forall d \neq d'; \forall c \neq c', \quad \mathbb{E}_{\boldsymbol{W}^{(l)}}[T_{dc}^{(l,uv)} T_{d'c'}^{(l,uv)}] = \mathbb{E}_{\boldsymbol{W}^{(l)}}[T_{dc}^{(l,uv)}] \mathbb{E}_{\boldsymbol{W}^{(l)}}[T_{d'c'}^{(l,uv)}]$$

$$\forall l, d, c, d', c', \quad \mathbb{E}_{W^{(l)},\dots,W^{(1)}}[T_{dc}^{(l,uv)} \mathbb{T}_{d'c'}^{(uv)(l-1:1)}] = \mathbb{E}_{W^{(l)}}[T_{dc}^{(l,uv)}] \mathbb{E}_{W^{(l-1)},\dots,W^{(1)}}[\mathbb{T}_{d'c'}^{(uv)(l-1:1)}]$$

*Besides, according to experimental experience, the mean value of all parameters in $W^{(l)}$ usually has a small bias during the training process, instead of being exactly zero. Therefore, let us assume that in early epochs, each parameter in $W^{(l)}$ is sampled from a Gaussian distribution $N(\mu_l, \sigma_l^2)$.*

*Proof.* Given an initialized, cascaded, convolutional decoder[1] network with $L$ convolutional layers, let us focus on the behavior of the decoder network in early epochs of training. We notice that each element in the matrix $T^{(l,uv)}$ is exclusively determined by the $c$-th channel of the $d$-th kenel $W_{c,1:K,1:K}^{(l)[ker=d]} \in \mathbb{R}^{K \times K}$ according to Equation (5). Because parameters in $W^{(l)}$ in the decoder network are set to random noises, we can consider that all elements in $T^{(l,uv)}$ irrelevant to each other, *i.e.*, $\forall d \neq d', c \neq c', T_{dc}^{(l,uv)}$ is irrelevant to $T_{d'c'}^{(l,uv)}$. Similarly, since different layers' parameters $W^{(l)}$ are irrelevant to each other in the initialized decoder network, we can consider that elements in different layers' $T^{(l,uv)}$ irrelevant to each other, *i.e.*, $\forall l \neq l'$, elements in $T^{(l,uv)}$ and elements in $T^{(l',uv)}$ are irrelevant to each other. Moreover, since the early training of a DNN mainly modifies a few parameters according to the lottery ticket hypothesis (Frankle & Carbin, 2018), we can still assume such irrelevant relationships in early epochs, as follows. $\quad\square$

Then, we prove Theorem 3.

*Theorem* 3. *Based on Assumption 1 and Assumption 2, we can prove that $T_{dc}^{(l,uv)}$ follows a Gaussian distribution of complex numbers, as follows.*

$$\forall d, c \quad T_{dc}^{(l,uv)} \sim ComplexN(\hat{\mu} = \mu_l R_{uv}, \hat{\sigma}^2 = K^2 \sigma_l^2, r = \sigma_l^2 R_{2u,2v})$$

$$s.t. \quad R_{uv} = \frac{\sin(uK\pi/M)}{\sin(u\pi/M)} \frac{\sin(vK\pi/N)}{\sin(v\pi/N)} e^{i(\frac{(K-1)u}{M} + \frac{(K-1)v}{N})\pi}$$

*Proof.* According to Assumption 2, each convolutional weight follows a Gaussian distribution, *i.e.*, $W_{cts}^{ker=d} \sim \mathcal{N}(\mu_l, \sigma_l^2)$. For the convenience of proving, let us extend $W_{cts}^{ker=d}$ into an complex number. In this way, $W_{cts}^{ker=d}$ follows a Gaussian distribution of complex numbers, *i.e.*, $W_{cts}^{ker=d} \sim ComplexN(\mu_l, \sigma_l^2, 0)$.

Previous studies Tse & Viswanath (2005) proved that given $N$ complex numbers, if each complex number follows a Gaussian distribution, then the linear summation of these $N$ complex numbers also follows a Gaussian distribution of complex numbers. Since $T_{dc}^{(l,uv)}$ is a linear combination of $\forall t, s, W_{cts}^{(l)[ker=d]}, T_{dc}^{(l,uv)}$ also follows a Gaussian distribution of complex numbers as follows.

$$\forall d, c \quad T_{dc}^{(l,uv)} \sim ComplexN(\hat{\mu}, \hat{\sigma}^2, r)$$

where

$$\mu = \mathbb{E}[T_{dc}^{(l,uv)}] \quad //\text{By definetion of } \mu$$

$$= \mathbb{E}[\sum_{t=0}^{K-1} \sum_{s=0}^{K-1} W_{cts}^{(l)[ker=d]} e^{i(\frac{ut}{M} + \frac{vs}{N})2\pi}] \quad //\text{Equation (11)}$$

$$//\forall t \neq t' \text{ or } s \neq s' : \mathbb{E}[W_{cts}^{(l)[ker=d]} W_{ct's'}^{(l)[ker=d]}] = \mathbb{E}[W_{cts}^{(l)[ker=d]}] \mathbb{E}[W_{ct's'}^{(l)[ker=d]}]$$

$$= \sum_{t=0}^{K-1} \sum_{s=0}^{K-1} \mathbb{E}[W_{cts}^{(l)[ker=d]}] e^{i(\frac{ut}{M} + \frac{vs}{N})2\pi}$$

$$= \mu_l \sum_{t=0}^{K-1} \sum_{s=0}^{K-1} e^{i(\frac{ut}{M} + \frac{vs}{N})2\pi} \quad //\mathbb{E}[W_{cts}^{(l)[ker=d]}] = \mu_l$$

$$//\text{let } R_{uv} = \sum_{t=0}^{K-1} \sum_{s=0}^{K-1} e^{i(\frac{ut}{M} + \frac{vs}{N})2\pi}$$

$$= \mu_l R_{uv}$$

$$\sigma^2 = \mathbb{E}[(T_{dc}^{(l,uv)} - \mathbb{E}[T_{dc}^{(l,uv)}])\overline{(T_{dc}^{(l,uv)} - \mathbb{E}[T_{dc}^{(l,uv)}])}] \quad //\text{By definetion of } \sigma^2$$

$$= Var[T_{dc}^{(l,uv)}]$$

$$= Var[\sum_{t=0}^{K-1}\sum_{s=0}^{K-1} W_{cts}^{(l)[\text{ker}=d]} e^{i(\frac{ut}{M}+\frac{vs}{N})2\pi}] \quad //\text{Equation (11)}$$

$$//\forall t \neq t' \text{ or } s \neq s' : \mathbb{E}[W_{cts}^{(l)[\text{ker=d}]}W_{ct's'}^{(l)[\text{ker=d}]}] = \mathbb{E}[W_{cts}^{(l)[\text{ker=d}]}]\mathbb{E}[W_{ct's'}^{(l)[\text{ker=d}]}]$$

$$= \sum_{t=0}^{K-1}\sum_{s=0}^{K-1} Var[W_{cts}^{(l)[\text{ker}=d]} e^{i(\frac{ut}{M}+\frac{vs}{N})2\pi}]$$

$$= \sum_{t=0}^{K-1}\sum_{s=0}^{K-1} Var[W_{cts}^{(l)[\text{ker}=d]}] \quad //Var[aX] = |a|^2 Var[X]$$

$$= \sum_{t=0}^{K-1}\sum_{s=0}^{K-1} \sigma_l^2 \quad //Var[W_{cts}^{(l)[\text{ker}=d]}] = \sigma_l^2$$

$$= K^2\sigma_l^2$$

$$r = \mathbb{E}[(T_{dc}^{(l,uv)} - \mathbb{E}[T_{dc}^{(l,uv)}])(T_{dc}^{(l,uv)} - \mathbb{E}[T_{dc}^{(l,uv)}])] \quad //\text{By definetion of r}$$

$$= C[T_{dc}^{(l,uv)}] \quad //\text{Define } C[\mathbf{X}] = \mathbb{E}[(\mathbf{X} - \mathbb{E}[\mathbf{X}])(\mathbf{X} - \mathbb{E}[\mathbf{X}])]$$

$$= C[\sum_{t=0}^{K-1}\sum_{s=0}^{K-1} W_{cts}^{(l)[\text{ker}=d]} e^{i(\frac{ut}{M}+\frac{vs}{N})2\pi}] \quad //\text{Equation (11)}$$

$$//\forall t \neq t' \text{ or } s \neq s' : \mathbb{E}[W_{cts}^{(l)[\text{ker=d}]}W_{ct's'}^{(l)[\text{ker=d}]}] = \mathbb{E}[W_{cts}^{(l)[\text{ker=d}]}]\mathbb{E}[W_{ct's'}^{(l)[\text{ker=d}]}]$$

$$= \sum_{t=0}^{K-1}\sum_{s=0}^{K-1} C[W_{cts}^{(l)[\text{ker}=d]} e^{i(\frac{ut}{M}+\frac{vs}{N})2\pi}]$$

$$= \sum_{t=0}^{K-1}\sum_{s=0}^{K-1} C[W_{cts}^{(l)[\text{ker}=d]}] e^{i(\frac{2ut}{M}+\frac{2vs}{N})2\pi} \quad //C[aX] = a^2 C[X]$$

$$= \sigma_l^2 \sum_{t=0}^{K-1}\sum_{s=0}^{K-1} e^{i(\frac{2ut}{M}+\frac{2vs}{N})2\pi} \quad //Var[W_{cts}^{(l)[\text{ker}=d]}] = \sigma_l^2$$

$$= \sigma_l^2 R_{2u,2v} \quad // R_{uv} = \sum_{t=0}^{K-1}\sum_{s=0}^{K-1} e^{i(\frac{ut}{M}+\frac{vs}{N})2\pi}$$

Finally, let us consider the value of $R_{uv}$.

$$R_{uv} = \sum_{t=0}^{K-1}\sum_{s=0}^{K-1} e^{i(\frac{ut}{M}+\frac{vs}{N})2\pi}$$

$$= \sum_{t=0}^{K-1} e^{i(\frac{2u\pi}{M})t} \sum_{s=0}^{K-1} e^{i(\frac{2v\pi}{N})s}$$

$$= \frac{\sin(\frac{Ku}{M}\pi)}{\sin(\frac{u}{M}\pi)} \cdot \frac{\sin(\frac{Kv}{N}\pi)}{\sin(\frac{v}{N}\pi)} \cdot e^{i(\frac{(K-1)u}{M}+\frac{(K-1)v}{N})\pi} \quad //\text{According to Equation (1)}$$

Therefore, we prove that

$$\forall d, c \quad T_{dc}^{(l,uv)} \sim Complex\mathcal{N}(\hat{\mu} = \mu_l R_{uv}, \hat{\sigma}^2 = K^2\sigma_l^2, r = \sigma_l^2 R_{2u,2v})$$

$$\text{s.t. } R_{uv} = \frac{\sin(uK\pi/M)}{\sin(u\pi/M)}\frac{\sin(vK\pi/N)}{\sin(v\pi/N)} e^{i(\frac{(K-1)u}{M}+\frac{(K-1)v}{N})\pi}$$

$\square$

### A.6 PROOF OF THEOREM 4

**Theorem 4.** (proven in Appendix A.5) *Based on Assumption 1 and Assumption 2, we can prove that* $T_{dc}^{(l,uv)}$ *follows a Gaussian distribution of complex numbers, as follows.*

$$\forall d, c \quad T_{dc}^{(l,uv)} \sim Complex\mathcal{N}(\hat{\mu} = \mu_l R_{uv}, \hat{\sigma}^2 = K^2\sigma_l^2, r = \sigma_l^2 R_{2u,2v})$$

$$s.t. \quad R_{uv} = \frac{\sin(uK\pi/M)}{\sin(u\pi/M)} \frac{\sin(vK\pi/N)}{\sin(v\pi/N)} e^{i(\frac{(K-1)u}{M} + \frac{(K-1)v}{N})\pi}$$

*Proof.* According to Theorem 4, $\forall d, c, l : \mathbb{E}[T_{dc}^{(l,uv)}] = \mu_l R_{uv}, Var[T_{dc}^{(l,uv)}] = K^2\sigma_l^2.$

$$
\begin{aligned}
SOM(T_{dc}^{(l,uv)}) &= \mathbb{E}[|T_{dc}^{(l,uv)}|^2] \\
&= |\mathbb{E}[T_{dc}^{(l,uv)}]|^2 + Var[T_{dc}^{(l,uv)}] \\
&= |\mu_l R_{uv}|^2 + K^2\sigma_l^2
\end{aligned}
\tag{19}
$$

Then, we have

$$
\begin{aligned}
\log(SOM(\mathbb{T}^{(uv)(L:1)})) &= \log(\mathbb{E}[|\mathbb{T}^{(uv)(L:1)}|^2]) \\
&= \log(\mathbb{E}[|T^{(L,uv)}\mathbb{T}^{(uv)(L-1:1)}|^2]) \\
&//\text{Assumption 2, and } C_l = 1 \\
&= \log(\mathbb{E}[|T^{(L,uv)}|^2]\mathbb{E}[|\mathbb{T}^{(uv)(L-1:1)}|^2]) \\
&= \log((|\mu_L R_{uv}|^2 + K^2\sigma_L^2)SOM(\mathbb{T}^{(uv)(L-1:1)})) \quad //\text{Equation (19)} \\
&= \log(\prod_{l=1}^{L} |\mu_l R_{uv}|^2 + K^2\sigma_l^2) \\
&= \sum_{l=1}^{L} \log(|\mu_l R_{uv}|^2 + K^2\sigma_l^2)
\end{aligned}
$$

$\square$

**For the more general case that each convolutional kernel contains more than one channel,** *i.e.*, $\forall l, C_l > 1$**, the** *SOM*$(\mathbb{T}^{(uv)(L:1)})$ **also approximately exponentially increases along with the depth of the network with a quite complicated analytic solution, as proved below.** Note that the following proof is based Assumption 2. Besides, we further assume that all element in $\mathbb{T}^{(uv)(l:1)}$ are independent with each other. *I.e.*, $\forall d \neq d'; c \neq c', \mathbb{E}[\mathbb{T}_{dc}^{(uv)(l:1)}\mathbb{T}_{d'c'}^{(uv)(l:1)}] = \mathbb{E}[\mathbb{T}_{dc}^{(uv)(l:1)}]\mathbb{E}[\mathbb{T}_{d'c'}^{(uv)(l:1)}].$

*Proof.* According to Theorem 4, all element in $T^{(l,uv)}$ follow the same Gaussian distribution. Therefore, we have

$$
\begin{aligned}
\mathbb{E}[T^{(l,uv)}] &= \mathbb{E}[T_{dc}^{(l,uv)}]\mathbf{1}_{(C_l \times C_{l-1})} \\
&= \mu_l R_{uv}\mathbf{1}_{(C_l \times C_{l-1})}
\end{aligned}
\tag{20}
$$

and we have

$$
\begin{aligned}
SOM(T^{(l,uv)}) &= SOM(T_{dc}^{(l,uv)})\mathbf{1}_{(C_l \times C_{l-1})} \\
&= (|\mu_l R_{uv}|^2 + K^2\sigma_l^2)\mathbf{1}_{(C_l \times C_{l-1})}
\end{aligned}
\tag{21}
$$

Let us first consider the expectation of $\mathbb{T}^{(uv)(L:1)}$ as follows.

$$\mathbb{E}[\mathbb{T}^{(uv)(L:1)}] = \mathbb{E}[T^{(L,uv)}\mathbb{T}^{(uv)(L-1:1)}]$$

$$= (C_{L-1}\mathbb{E}[T_{dc}^{(L,uv)}]\mathbb{E}[\mathbb{T}_{dc}^{(uv)(L-1:1)}])\mathbf{1}_{(C_L \times C_0)} \quad //\text{Assumption 2, Equation (20)}$$

$$= (C_{L-1}\mu_l R_{uv}\mathbb{E}[\mathbb{T}_{dc}^{(uv)(L-1:1)}])\mathbf{1}_{(C_L \times C_0)} \quad //\text{Theorem 4}$$

$$= (\frac{1}{C_L}\prod_{l=1}^{L}C_l\mu_l R_{uv})\mathbf{1}_{(C_L \times C_0)} \quad //\text{Assumption 2}$$

$$\tag{22}$$

Then, we have

$$SOM(\mathbb{T}^{(uv)(L:1)})$$

$$= \mathbb{E}[|\mathbb{T}^{(uv)(L:1)}|^2]$$

$$= \mathbb{E}[|T^{(L,uv)}\mathbb{T}^{(uv)(L-1:1)}|^2]$$

$$= (C_{L-1}SOM(T_{dc}^{(L,uv)})SOM(\mathbb{T}_{dc}^{(uv)(L-1:1)}) + C_{L-1}(C_{L-1} - 1)|\mathbb{E}[T_{dc}^{(L,uv)}]\mathbb{E}[\mathbb{T}_{dc}^{(uv)(L-1:1)}]|^2)\mathbf{1}_{(C_L \times C_0)}$$

$$//\text{According to Assumption 2 and Equation (21)},$$

$$//\text{we further Assume } \forall d \neq d'; c \neq c', \mathbb{E}[\mathbb{T}_{dc}^{(uv)(l:1)}\mathbb{T}_{d'c'}^{(uv)(l:1)}] = \mathbb{E}[\mathbb{T}_{dc}^{(uv)(l:1)}]\mathbb{E}[\mathbb{T}_{d'c'}^{(uv)(l:1)}]$$

$$= (C_{L-1}(|\mu_L R_{uv}|^2 + K^2\sigma_L^2)SOM(\mathbb{T}_{dc}^{(uv)(L-1:1)}) + \frac{C_{L-1} - 1}{C_{L-1}}|\mathbb{E}[\mathbb{T}_{dc}^{(uv)(L:1)}]|^2)\mathbf{1}_{(C_L \times C_0)}$$

$$//\text{According to Equation (19), Equation (22)}$$

$$= \left(\frac{1}{C_L}\prod_{l=1}^{L}C_l(|\mu_l R_{u,v}|^2 + (K\sigma_l)^2) + \sum_{l=2}^{L}\frac{C_{l-1} - 1}{C_{l-1}}|\frac{1}{C_l}\prod_{k=1}^{l}C_k\mu_k R_{u,v}|^2\prod_{j=l+1}^{L}C_{j-1}\left(|\mu_j R_{u,v}|^2\right.\right.$$

$$\left.\left. + (K\sigma_j)^2\right)\right)\mathbf{1}_{C_L \times C_0}$$

$$\tag{23}$$

Therefore, we prove that for the more general case that $\forall l, C_l > 1$, the second-order moment $SOM(\mathbb{T}^{(uv)(L:1)})$ also approximately exponentially increases along with the depth of the network.

$\square$

## A.7 PROOF OF THEOREM 5

In this section, we prove Theorem 5 in the main paper.

*Theorem 5. Let each element in each c-th channel $F^{(c)}$ of the feature map follows the Gaussian distribution $\mathcal{N}(a, \sigma^2)$. $G^{(c)} \in \mathbb{C}^{M \times N}$ denotes the frequency spectrum of $F^{(c)}$, and $H^{(c)} \in \mathbb{C}^{M' \times N'}$ denotes the frequency spectrum of the output feature $\widetilde{F}^{(c)}$ after applying zero-padding on $F^{(c)}$. Then, the zero-padding on $F^{(c)}$ brings in additional signals at each frequency $[u, v]$ as follows, whose strength is measured by averaging over different sampled features.*

$$\forall 0 \leq u < M, \ 0 \leq v < N, \ \mathbb{E}_{F^{(c)}}[|H_{uv}^{(c)} - G_{uv}^{(c)}|] = |a|\left(\left|\frac{\sin(Mu\pi/M')}{\sin(u\pi/M')}\frac{Nv\pi/N'}{v\pi/N'}\right| - MN\delta_{uv}\right);$$

$$\forall M \leq u < M', \ N \leq v < N', \ \mathbb{E}_{F^{(c)}}[|H_{uv}^{(c)}|] = \left|a\frac{\sin(Mu\pi/M')}{\sin(u\pi/M')}\frac{Nv\pi/N'}{v\pi/N'}e^{-i(\frac{(M-1)u}{M'} + \frac{(N-1)v}{N'})\pi}\right|$$

*Proof.*

$$\mathbb{E}_{F^{(c)}}[G_{uv}^{(c)}] = \mathbb{E}[\sum_{m=0}^{M-1}\sum_{n=0}^{N-1} F_{mn}^{(c)} e^{-i(\frac{um}{M}+\frac{vn}{N})2\pi}] \quad //\text{Equation (11)}$$

$$= \sum_{m=0}^{M-1}\sum_{n=0}^{N-1} \mathbb{E}[F_{mn}^{(c)}] e^{-i(\frac{um}{M}+\frac{vn}{N})2\pi} \tag{24}$$

$$= a\sum_{m=0}^{M-1}\sum_{n=0}^{N-1} e^{-i(\frac{um}{M}+\frac{vn}{N})2\pi} \quad //\text{F}_{mn}^{(c)} \sim \mathcal{N}(a,\sigma^2)$$

$$= aMN\delta_{uv}; 0 \le u < M, 0 \le v < N \quad //\text{Equation (3)}$$

$$\mathbb{E}_{F^{(c)}}[H_{uv}^{(c)}]$$

$$= \mathbb{E}_{F^{(c)}}[\sum_{m=0}^{M'-1}\sum_{n=0}^{N'-1} \tilde{F}_{mn}^{(c)} e^{-i(\frac{um}{M'}+\frac{vn}{N'})2\pi}] \quad //\text{Equation (11)}$$

$$= \mathbb{E}_{F^{(c)}}[\sum_{m=0}^{M-1}\sum_{n=0}^{N-1} F_{mn}^{(c)} e^{-i(\frac{um}{M'}+\frac{vn}{N'})2\pi}]$$

$$= a\sum_{m=0}^{M-1}\sum_{n=0}^{N-1} e^{-i(\frac{um}{M'}+\frac{vn}{N'})2\pi} \quad //\text{F}_{mn}^{(c)} \sim \mathcal{N}(a,\sigma^2)$$

$$= a\frac{\sin(\frac{Mu}{M'}\pi)}{\sin(\frac{u}{M'}\pi)}\frac{\sin(\frac{Nv}{N'}\pi)}{\sin(\frac{v}{N'}\pi)} e^{-i(\frac{(M-1)u}{M'}+\frac{(N-1)v}{N'})\pi}; 0 \le u < M', 0 \le v < N' \quad //\text{Equation (1)}$$

$$\tag{25}$$

When $0 \le u < M, 0 \le v < N$

$$\mathbb{E}_{F^{(c)}}[|H_{uv}^{(c)} - G_{uv}^{(c)}|] = |\mathbb{E}_{F^{(c)}}[H_{uv}^{(c)} - G_{uv}^{(c)}]|$$

$$= |\mathbb{E}_{F^{(c)}}[H_{uv}^{(c)}] - \mathbb{E}_{F^{(c)}}[G_{uv}^{(c)}]|$$

$$//\text{According to Equation 24, Equation 25}$$

$$= |a\frac{\sin(\frac{Mu}{M'}\pi)}{\sin(\frac{u}{M'}\pi)}\frac{\sin(\frac{Nv}{N'}\pi)}{\sin(\frac{v}{N'}\pi)} e^{-i(\frac{(M-1)u}{M'}+\frac{(N-1)v}{N'})\pi} - aMN\delta_{uv}|$$

$$= |a|(|\frac{\sin(\frac{Mu}{M'}\pi)}{\sin(\frac{u}{M'}\pi)}\frac{\sin(\frac{Nv}{N'}\pi)}{\sin(\frac{v}{N'}\pi)}| - MN\delta_{uv})$$

When $M \le u < M', N \le v < N'$

$$\mathbb{E}_{F^{(c)}}[|H_{uv}^{(c)}|] = |\mathbb{E}_{F^{(c)}}[H_{uv}^{(c)}]|$$

$$= |a\frac{\sin(\frac{Mu}{M'}\pi)}{\sin(\frac{u}{M'}\pi)}\frac{\sin(\frac{Nv}{N'}\pi)}{\sin(\frac{v}{N'}\pi)} e^{-i(\frac{(M-1)u}{M'}+\frac{(N-1)v}{N'})\pi}| \quad //\text{Equation (25)}$$

$$= |a||\frac{\sin(\frac{Mu}{M'}\pi)}{\sin(\frac{u}{M'}\pi)}\frac{\sin(\frac{Nv}{N'}\pi)}{\sin(\frac{v}{N'}\pi)}|$$

Therefore, we prove that

$$\forall 0 < u < M, 0 < c < N, \mathbb{E}_{F^{(c)}}[|H_{uv}^c - G_{uv}^c|] = |a|(|\frac{\sin(\frac{Mu}{M'}\pi)}{\sin(\frac{u}{M'}\pi)}\frac{\sin(\frac{Nv}{N'}\pi)}{\sin(\frac{v}{N'}\pi)}| - MN\delta_{uv})$$

$$\forall M < u < M', N < c < N', \mathbb{E}_{F^{(c)}}[|H_{uv}^c|] = |a\frac{\sin(\frac{Mu}{M'}\pi)}{\sin(\frac{u}{M'}\pi)}\frac{\sin(\frac{Nv}{N'}\pi)}{\sin(\frac{v}{N'}\pi)}|$$

$$\square$$

## A.8    PROOF OF THEOREM 6

In this section, we prove Theorem 6 in the main paper.

**Theorem 6.** *Let $\boldsymbol{G} = [G^{(1)}, G^{(2)}, \ldots, G^{(C_l)}] \in \mathbb{C}^{C_l \times M_0 \times N_0}$ denote spectrums of the $C_l$ channels of feature $\boldsymbol{F}$. Then, spectrums $\boldsymbol{H} = [H^{(1)}, H^{(2)}, \ldots, H^{(C_l)}] \in \mathbb{C}^{C_l \times M \times N}$ of the output feature $\widetilde{\boldsymbol{F}}$ can be computed as follows.*

$$\forall c, u, v, \quad H^{(c)}_{u+(s-1)M_0, v+(t-1)N_0} = G^{(c)}_{uv} \qquad \text{s.t. } s = 1, \ldots, M/M_0; \; t = 1, \ldots, N/N_0 \qquad (26)$$

*Proof.*

$$G^{(c)}_{uv} = \sum_{m=0}^{M_0-1} \sum_{n=0}^{N_0-1} F^{(c)}_{mn} e^{-i(\frac{um}{M_0} + \frac{vn}{N_0})2\pi} \quad //\text{Equation (11)} \qquad (27)$$

$$
\begin{aligned}
H^{(c)}_{u+(s-1)M_0, v+(t-1)N_0} &= \sum_{m=0}^{M-1} \sum_{n=0}^{N-1} \tilde{F}^{(c)}_{mn} e^{-i(\frac{(u+(s-1)M_0)m}{M} + \frac{(v+(t-1)N_0)n}{N})2\pi} \quad //\text{Equation (11)} \\
&= \sum_{m=0}^{M_0-1} \sum_{n=0}^{N_0-1} F^{(c)}_{mn} e^{-i(\frac{(u+(s-1)M_0)(m \cdot ratio)}{M} + \frac{(v+(t-1)N_0)(n \cdot ratio)}{N})2\pi} \\
&= \sum_{m=0}^{M_0-1} \sum_{n=0}^{N_0-1} F^{(c)}_{mn} e^{-i(\frac{(u+(s-1)M_0)m}{M/ratio} + \frac{(v+(t-1)N_0)n}{N/ratio})2\pi} \\
&\quad //M = M_0 \cdot ratio; N = N_0 \cdot ratio \\
&= \sum_{m=0}^{M_0-1} \sum_{n=0}^{N_0-1} F^{(c)}_{mn} e^{-i(\frac{(u+(s-1)M_0)m}{M_0} + \frac{(v+(t-1)N_0)n}{N_0})2\pi} \\
&= \sum_{m=0}^{M_0-1} \sum_{n=0}^{N_0-1} F^{(c)}_{mn} e^{-i(\frac{um}{M_0} + \frac{vn}{N_0})2\pi} \cdot e^{-i((s-1)m + (t-1)n)2\pi} \\
&= \sum_{m=0}^{M_0-1} \sum_{n=0}^{N_0-1} F^{(c)}_{mn} e^{-i(\frac{um}{M_0} + \frac{vn}{N_0})2\pi} \quad //s, t \in \mathcal{Z} \\
&= G^{(c)}_{uv} \quad //\text{Equation (27)}
\end{aligned}
\qquad (28)
$$

Therefore we prove that:

$$\forall c, u, v, \quad H^{(c)}_{u+(s-1)M_0, v+(t-1)N_0} = G^{(c)}_{uv} \qquad \text{s.t. } s = 1, \ldots, M/M_0; \; t = 1, \ldots, N/N_0 \qquad (29)$$

$\square$

# B    RELATED WORK

Although few previous studies directly prove a DNN's bottleneck from the perspective of representing specific feature components, we still make a survey on research on the representation capacity of a DNN.

Some studies focused on a specific frequency that took the landscape of the loss function on all input samples as the time domain (Xu et al., 2019b; Rahaman et al., 2019; Xu et al., 2019a; Luo et al., 2019). Based on such a specific frequency, they observed and proved a phenomenon namely Frequency Principle (F-Principle) that a DNN first qucikly learned low-frequency components, and then relatively slowly learned the high-frequency ones, which might shed new light on understanding the representation capacity of a DNN. For example, Lin et al. (2019) empirically proposed to smooth out high-frequency components to improve the adversarial robustness. Besides, Ma et al. (2020) explored the boundary of the F-Principle, beyond which the F-Principle did not hold any more. **In comparison, we focus on a fully different type of frequency**, *i.e.*, the frequency *w.r.t.* the DFT on an input image or a feature map.

In this direction, previous studies mainly experimentally analyzed the relationship between the learning of different frequencies and the robustness of a DNN. Yin et al. (2019) conducted a lot of experiments to analyze the robustness of a DNN *w.r.t.* different frequencies of the image. They discovered that both adversarial training and Gaussian data augmentation improved the DNN's robustness to higher frequencies. Wang et al. (2020) empirically proposed to remove high frequency components of convolutional weights to improve the adversarial robustness. In comparison, we theoretically prove representations bottleneck of DNNs in the frequency domain.

Besides, many studies explained the representation capacity of a DNN in the time domain. The information bottleneck hypothesis shows that the learning process of DNNs is to retain the task-relevant input information and discard the task-irrelevant input information (Tishby & Zaslavsky, 2015; Shwartz-Ziv & Tishby, 2017; Wolchover & Reading, 2017; Amjad & Geiger, 2019). The lottery ticket hypothesis shows that some initial parameters of DNNs inherently contribute more to the network output (Frankle & Carbin, 2018). The double-descent phenomenon describes the specific training process of DNNs that the loss first declines, then rises, and then declines again (Nakkiran et al., 2019; Reinhard & Fatih, 2020). DNNs with the batch normalization may sometimes conflicted with the weight decay (Van Laarhoven, 2017; Li et al., 2020). DNNs are vulnerable to adversarial examples (Szegedy et al., 2013; Goodfellow et al., 2014). DNNs tipically encoded simple interactions between very few input variables and complex interactions between almost all input variables, but were difficult to encode interactions between intermediate number of input variables (Deng et al., 2022).

## C   MORE EXPERIMENTAL RESULTS

### C.1   VERIFYING THAT A NEURAL NETWORK USUALLY LEARNED LOW-FREQUENT COMPONENTS FIRST.

In section, we provide more experimental results to verify that a neural network usually learned low-frequent components first, which had already been shown in Figure 1(a) in the main paper. Here, we also constructed a cascaded convolutional auto-encoder by using the VGG-16 as the encoder network. The decoder network contained three upconvolutional layers for the CIFAR-10 dataset, and contained three upconvolutional layers for the Broden dataset. Each convolutional/upconvolutional layer in the auto-encoder applied zero-paddings and was followed by a batch normalization layer and an ReLU layer. The auto-encoder was trained using the mean squared error (MSE) loss for image reconstruction. Results in Figure 6 verified that the auto-encoder usually learned low-frequent components first and gradually learned higher frequecies. We also attached the generated image below its spectrum map in Figure 7, in order to help people understand the learning process of the auto-encoder.

### C.2   VERIFYING THAT THE UPSAMPLING OPERATION MADE A DECODER NETWORK REPEAT STRONG SIGNALS AT CERTAIN FREQUENCIES OF THE GENERATED IMAGE.

In section, we provide more experimental results to verify that the upsampling operation in the decoder repeats strong frequency components of the input to generate spectrums of upper layers.

First, we conducted experiments to verify Theorem 6 in the main paper, which claims that the upsampling operation repeats the strong magnitude of the fundamental frequency $G_{00}^{(c)}$ of the lower layer to different frequency components $\forall c, H_{u^*v^*}^{(c)}$ of the higher layer, where $u^* = 0, M_0, 2M_0, 3M_0, \ldots; v^* = 0, N_0, 2N_0, 3N_0, \ldots$. To verify this, given an image, let the image pass through four cascaded upsampling layers. We visualized the feature spectrum generated by each upsampling layer, in order to verify whether the upsampling operation repeated the strong magnitude of the fundamental frequency of the input image to different frequency components of the feature spectrum generated by upsampling layers. Results on the CIFAR-10 dataset and the Tiny-ImageNet dataset in Figure 8 verified Theorem 6.

Second, we provide more results on real neural networks, which have already been shown in Figure 1(b) in the main paper. We also constructed a cascaded convolutional auto-encoder by using the VGG-16 as the encoder network. The decoder network contained four upconvolutional layers. Each convolutional/upconvolutional layer in the auto-encoder applied zero-paddings and was followed by

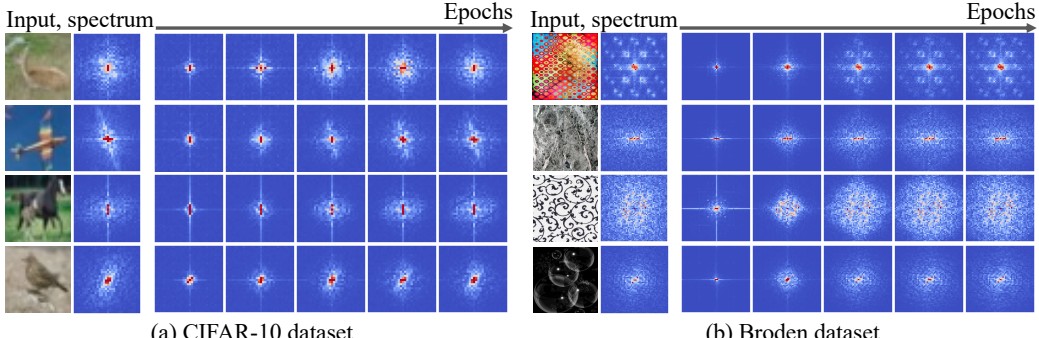

Figure 6: Magnitude maps of feature spectrums of different epochs' network output. Each magnitude map was averaged over all channels. For clarity, we moved low frequencies to the center of the spectrum map, and moved high frequencies to corners of the spectrum map. Note that we set the magnitude of the fundamental frequency to be the same with the frequency that had the second large magnitude. For resutls in (b), we only visualized components in the center of the spectrum map with the range of relatively low frequencies $u \in \{u|0 \leq u < M/8\} \cup \{u|7M/8 \leq u < M\}; v \in \{v|0 \leq v < N/8\} \cup \{v|7N/8 \leq v < N\}$ for clarity.

a batch normalization layer and an ReLU layer. The auto-encoder was trained on the Broden dataset using the mean squared error (MSE) loss for image reconstruction. Results in Figure 9 verified Theorem 6.

## C.3 VERIFYING THAT THE ZERO-PADDING OPERATION STRENGTHENED THE ENCODING OF LOW-FREQUENCY COMPONENTS.

In section, we provide more experimental results to verify that the zero-padding operation strengthened the encoding of low-frequency components, which had already been shown in Figure 5(c) in the main paper. Here, we also constructed the following two baseline networks. The first baseline network contained 5 convolutional layers, and each layer applied zero-paddings. Each convolutional layer contained 16 convolutional kernels (kernel size was $7 \times 7$), except for the last layer containing 3 convolutional kernels. The second baseline network was constructed by replacing all zero-padding operations with circular padding operations. Results in Figure 10 verified that the zero-padding operation strengthened the encoding of low-frequency components.

## C.4 VERIFYING THAT A DEEP NETWORK STRENGTHENED LOW-FREQUENCY COMPONENTS.

In section, we provide more experimental results to verify that a deep network strengthened low-frequency components, which had already been shown in Figure 5(a) in the main paper. Here, we also constructed a network with 50 convolutional layers. Each convolutional layer applied zero-paddings to avoid changing the size of feature maps, and was followed by an ReLU layer. We visualized feature spectrums of different convolutional layers. Results on the CIFAR-10 dataset and the Tiny-ImageNet dataset in Figure 11 show that magnitudes of low-frequency components increased along with the network layer number.

## C.5 VERIFYING THAT A LARGER ABSOLUTE MEAN VALUE $\mu_l$ OF EACH $l$-TH LAYER'S PARAMETERS STRENGTHENED LOW-FREQUENCY COMPONENTS.

In section, we provide more experimental results to verify that a larger absolute mean value $\mu_l$ of each $l$-th layer's parameters strengthened low-frequency components, which had already been shown in Figure 5(b) in the main paper. Here, we also applied a network architecture with 5 convolutional layers. Each layer contained 16 convolutional kernels (kernel size was $9 \times 9$), except for the last layer containing 3 convolutional kernels. Based on this architecture, we constructed three networks, whose parameters were sampled from Gaussian distributions $\mathcal{N}(\mu = 0, \sigma^2 = 0.01^2)$, $\mathcal{N}(\mu = 0.001, \sigma^2 = 0.01^2)$, and $\mathcal{N}(\mu = 0.01, \sigma^2 = 0.01^2)$, respectively. Results on the CIFAR-10

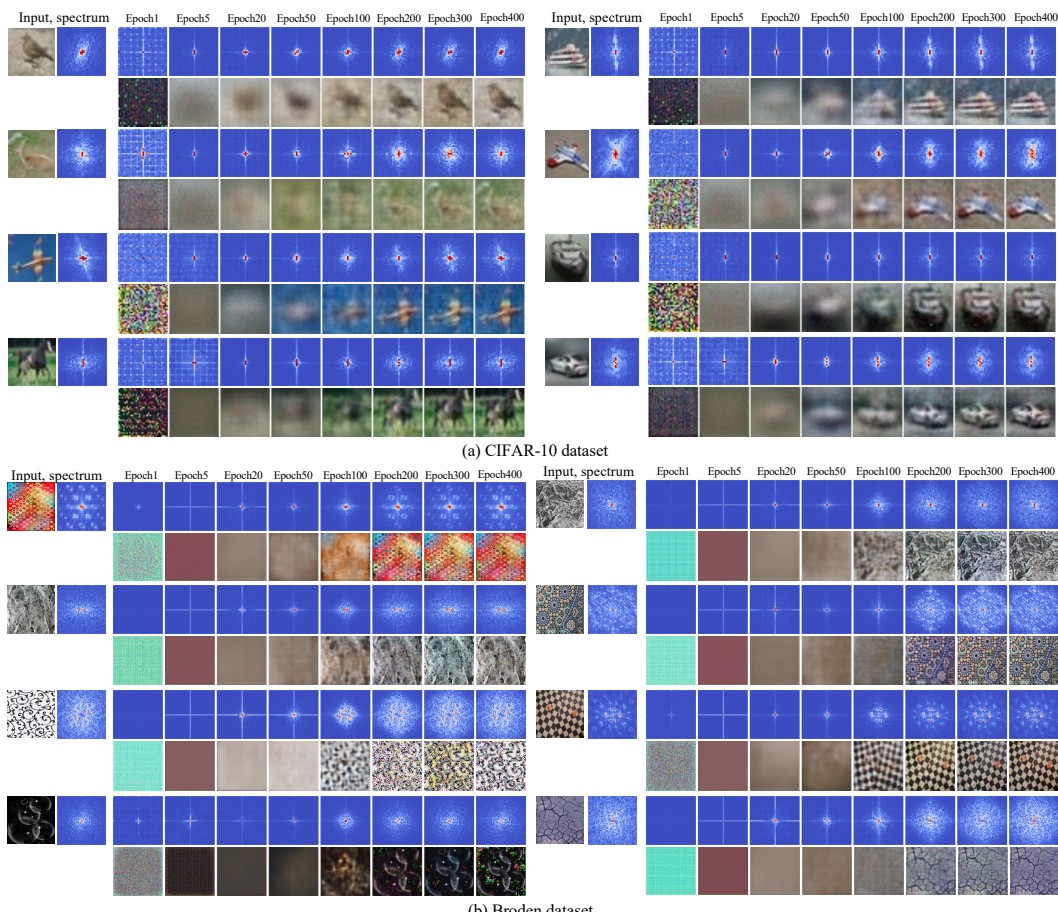

Figure 7: Magnitude maps of feature spectrums and the corresponding generated images of different epochs. Results show that in the very few epochs of the training, the network removed noisy signal caused by the upsampling, to some extent, which were in the grid pattern in the spectrum. After that, the network learned low-frequency components first, and then gradually learned higher frequencies. Each magnitude map in this figure was averaged over all channels. For clarity, we moved low frequencies to the center of the spectrum map, and moved high frequencies to corners of the spectrum map. Note that we set the magnitude of the fundamental frequency to be the same with the frequency that had the second large magnitude.

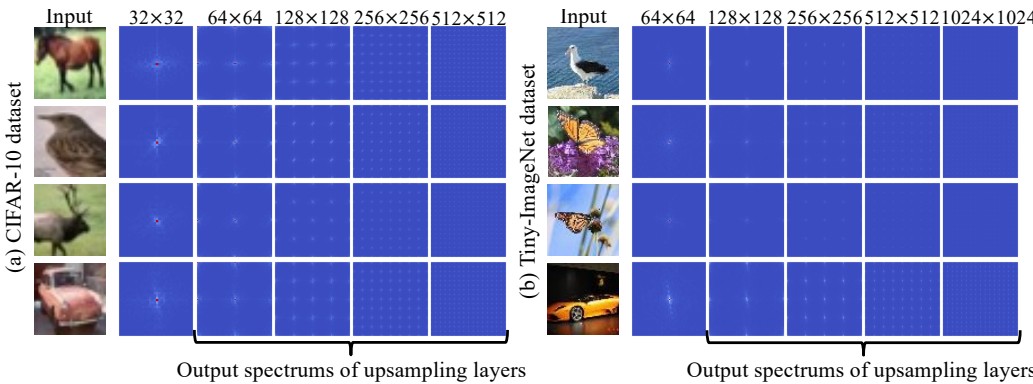

Figure 8: Magnitude maps of feature spectrums after one/two/there/four upsampling layers. Each magnitude map was averaged over all channels. For clarity, we moved low frequencies to the center of the spectrum map, and moved high frequencies to corners of the spectrum map.

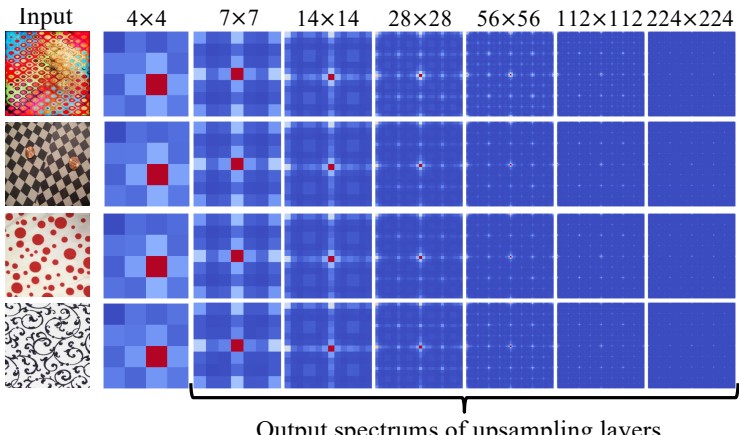

Figure 9: Magnitude maps of feature spectrums after one/two/there/four/five/six upsampling layers. Each magnitude map was averaged over all channels. For clarity, we moved low frequencies to the center of the spectrum map, and moved high frequencies to corners of the spectrum map.

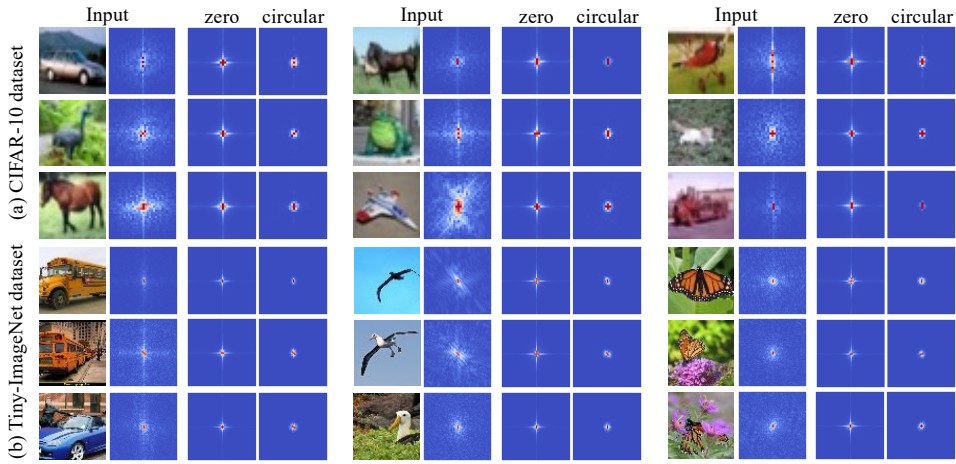

Figure 10: A network with zero-padding operations usually strengthened more low-frequency components than a network with circular padding operations. Here, each magnitude map of the feature spectrum was averaged over all channels. For clarity, we move low frequencies to the center of the spectrum map, move high frequencies to corners of the spectrum map, and set the magnitude of the fundamental frequency to be the same with the frequency that has the second large magnitude.

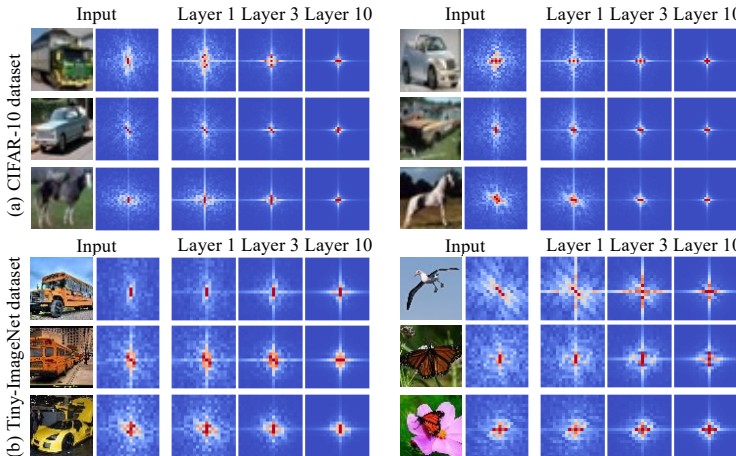

Figure 11: Comparing feature spectrums of different layers. Results show that higher layers of a network usually generated features with more low-frequency components. For clarity, we move low frequencies to the center of the spectrum map, move high frequencies to corners of the spectrum map, and set the magnitude of the fundamental frequency to be the same with the frequency that has the second large magnitude. For resutls in (b), we only visualized components in the center of the spectrum map with the range of relatively low frequencies $u \in \{u|0 \leq u < M/6\} \cup \{u|5M/6 \leq u < M\}; v \in \{v|0 \leq v < N/6\} \cup \{v|5N/6 \leq v < N\}$ for clarity.

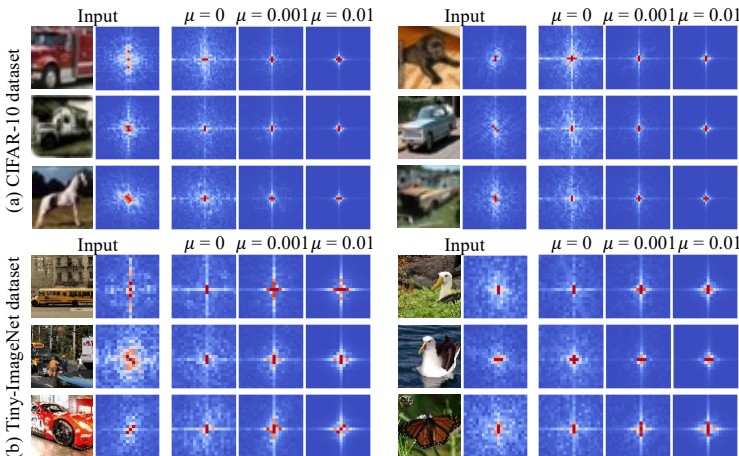

Figure 12: A network whose convolutional weights had a mean value significantly biased from 0 usually strengthened low-frequency components, but weakened high-frequency components. Here, each magnitude map of the feature spectrum was averaged over all channels. For clarity, we moved low frequencies to the center of the spectrum map, moved high frequencies to corners of the spectrum map, and set the magnitude of the fundamental frequency to be the same with the frequency that has the second large magnitude. For resutls in (b), we only visualized components in the center of the spectrum map with the range of relatively low frequencies $u \in \{u|0 \leq u < M/6\} \cup \{u|5M/6 \leq u < M\}; v \in \{v|0 \leq v < N/6\} \cup \{v|5N/6 \leq v < N\}$ for clarity.

dataset and the Tiny-ImageNet dataset in Figure 12 show that magnitudes of low-frequency components increased along with the absolute mean value of parameters.

