# OpenReview forum: "Explaining Representation Bottlenecks of Convolutional Decoder Networks"
_ICLR.cc/2023/Conference — Submitted to ICLR 2023_

### Official Review · Reviewer_fHYG · 2022-10-12

**Confidence:** 4
**Correctness:** 2
**Technical Novelty And Significance:** 1
**Empirical Novelty And Significance:** 1
**Recommendation:** 3

**Clarity, Quality, Novelty And Reproducibility:**

The novelty of the paper is not clear. The formulated model for ConvNets has oversimplified assumptions (removing non-linearities) and some the presented claims seams to be presented in well known theory (Fourier support of Convolutions) or previous papers on spectral bias.

**Strength And Weaknesses:**

The formulation seams to claim as novel existent well known concepts, at the same time that oversimplify the analytical model of ConvNets.



**Summary Of The Paper:**

The paper claims to prove representation bottlenecks of ConvNets considering the capacity of representing different frequency components of an input sample. It claims to introduce the rule of the forward propagation of intermediate-layer spectrum maps, as equivalent to the forward propagation of feature maps through a convolutional layer. It focus on layers representing convolutions, and zero-padding operations.


**Summary Of The Review:**

I dont see the novelty of this paper.

It presents as new, a theorem that conducting the convolution operation on an input feature F is essentially equivalent to
conducting matrix multiplication on spectrums of F (Theo 2). That is a well known results for Fourier analysis.

Next, it reformulates the cascade of convolutional layers, without activation functions. It is also well known that a stack of convolutional layers can be rewritten as a single convolutional layer if no non-linearities are used. But such a simplification is much less expressive than deep models adopting non-linearities.

On the experimental section, the paper claims to 'Verifying that a neural network usually learned low-frequent components first.' This result and analysis was previously presented in Rahaman et al.(2018) On the Spectral Bias of Neural Networks.

The paper also investigate the effect of zero padding, and upsampler layers, but for this last one, the formulation adopted seams to be the introduction of zeros on new subpixel positions (similar to those modeled by a dilated convolutions), other than investigating what happen with upsampling filters.

---

### Official Review · Reviewer_V1PA · 2022-10-22

**Confidence:** 3
**Correctness:** 2
**Technical Novelty And Significance:** 2
**Empirical Novelty And Significance:** 2
**Recommendation:** 3

**Clarity, Quality, Novelty And Reproducibility:**

The writing clarity and quality is ok. There are many minor grammar mistakes though. There seems to be novelty in the works, and it should be reproducible.


**Strength And Weaknesses:**

Strengths:

—-------------------

The paper has many strengths and is tackling an important problem. For the sake of time  I am only writing about weaknesses in this review, since those are the ones that should be actioned upon.


Weaknesses (W)

—-------------------

W: As far as I understand the analysis for the deep network is done on an idealized frequency domain network that does not correspond to a CNN with ReLUs and finite size filters. If this is true, this is the biggest drawback of the current paper. Furthermore, in real world networks, very small filters (e.g., 3x3 or even 2x2) usually provide the best results.

W: Keeping in mind the previous comment, the paper’s language might be too strong. It talks about proving various aspects of the forward propagation of the CNN, but maybe the proofs only apply to the idealized network?

W: Regarding zero-padding, there are previous works that try to compensate for the resulting border effects, e.g., by learning different filters to the borders or using circular or other types of padding and convolutions. Could the authors discuss how zero padding vs. these methods affect the end-results of the CNN decoder and whether it would be beneficial not to zero-pad.

W: The paper makes the claim that decoder CNN is biased towards learning low-frequency content. What is missing is the analysis on when this is actually harmful and whether it could be beneficial in some settings. Also, in cases where this is harmful, it would be interesting if the authors could provide a suggestion for improvement.

W: The analysis is done for randomly initialized networks. While this is an interesting point of analysis, it is lacking the effect of training altogether. I would suggest the authors to extend their work towards the training as well.

W: Based on Figure 3, ReLU seems to have a big effect when comparing the freq domain version of the network with the original one. A big downside of the analysis is the lack of analysing the effect of ReLU, since, of course, the activation function is a key component in enabling nonlinearity in the neural networks.

W: The authors should bring the related work section from the appendix to the main paper and also build a connection with related works on freq domain neural networks, for example. Pan, H., Chen, Y., Niu, X., Zhou, W., and Li, D., “Learning Convolutional Neural Networks in the Frequency Domain”, arXiv e-prints, 2022. https://arxiv.org/abs/2204.06718 , and many others.

W: The authors note that their work is different from the “F-Principle” e.g., https://arxiv.org/abs/1807.01251 because the current paper focuses on a “fully different type of frequency”, i.e., the frequency w.r.t. the DFT on an input image or a feature map. To me this distinction is not so clear and both works seem to be related. I would suggest the authors to build a better bridge between the works, since even the authors seem to be doing similar experiments, e.g., in “C.1 VERIFYING THAT A NEURAL NETWORK USUALLY LEARNED LOW-FREQUENT COMPONENTS FIRST.”



Minor issues and spelling mistakes:

B, page 26: “DNN first qucikly” -> DNN first quickly

Page 27: “DNNs tipically” -> DNNs typically



**Summary Of The Paper:**

This paper studies the forward pass of a convolutional decoder in the frequency domain. It finds that the CNN layer forward propagates each frequency component in the
spectrum map independently to other frequency components.
It also finds that the CNN operations make a convolutional decoder network more likely to weaken high-frequency components.


**Summary Of The Review:**

Decent work with some issues that would warrant a new revision before acceptance.

---

### Official Review · Reviewer_wmw7 · 2022-10-26

**Confidence:** 4
**Correctness:** 3
**Technical Novelty And Significance:** 2
**Empirical Novelty And Significance:** 1
**Recommendation:** 3

**Clarity, Quality, Novelty And Reproducibility:**

The paper is clear and nicely written. Almost all the findings of this paper are known basic principles of convolutions in signal processing. Hence, the work has minimal new findings.

**Strength And Weaknesses:**

The paper is written well and explained very clearly. The authors make it clear how their approach to frequency encoding differs from prior works. I very much enjoyed reading the paper. However, I have mixed feelings about this work. The paper gives lots of insights into linear convolutional neural networks. For someone with a signal processing background, however, I find the work to explain the known basics of discrete-time signal processing [1] to the deep learning community. Below I explain in detail.

I found the work to miss acknowledging the digital signal processing literature and their terminology in this paper. The paper misses the literature on discrete-time signal processing. All the properties mentioned in this paper are known for convolution operators. Moreover, the authors lack certain terminologies known in the literature. For example, the "repeats strong frequency components of the input" in Figure 1 and (Conclusion 4) is known as "aliasing". Below are a few examples.
- Their first theorem and the second are basic principles of convolutions and discrete Fourier transform. Could the authors explain what is new about this result?
- The independence of DFT points in a convolution operator is known in the literature. What is new here?
- Upsampling and aliasing are known in the signal processing literature. It is not clear what is new in their results on upsampling.
- It is very clear from the perspective of signal processing that if the kernels are initialized with Gaussian distributed data, then the spectrum allows follows a Gaussian shape, and depending on the variance, etc. it acts as smoothing. Is this something new?

There are certain simplifications that are extreme.
- On page 3, the authors mention that the diffraction process is small and can be ignored, but perhaps the main reason the cosine similarity in Figure 3 is not exactly 1 or very very close to 1 is due to the diffraction process term. Assuming the theorem is correct, Figure 3 indicates that the diffraction process should not be ignored.
- Non-linear activations are building blocks of neural networks. This paper considers a linear network. This is extreme simplification and reduces the setting into a case that is known and studied already in the literature (Convolution operator or cascade of convolution operators are known for how they perform in the signal processing literature). Please state clearly in the abstract that you are considering a linear network.
- They provide analysis of the network when it is NOT trained (or very early stage which is still assumed to behave like an untrained network) with independent variables and the assumption that the parameters are drawn from Gaussian distribution.

Misleading statements:
- "zero-padding operation boosts magnitudes of low-frequency components of feature spectrums of the feature map". Zero-padding in the spatial-image domain, from the perspective of DFT, increases the resolution in the frequency domain. It does not boost low-freq or filter high.
- The author's statement on the weakening of high-freq encoding is not generally true. This depends on the frequencies that each convolutional kernel passes. In their framework, they assume that the kernels pass low frequencies implicitly by using Gaussian distributed data. Moreover, this depends on the data the network will be trained on. For example, as pointed out by the authors in Remark 5, natural images dominate low-frequencies, hence, CNNs trained on natural images tend to filter high-frequencies.

Some word usage is not precise: "Random noises" in Section 3? I guess you mean drawn randomly from a particular (Gaussian) distribution. Please clarify "irrelevant" in Section 3? Do you mean "independent"? By "upsampling" do you mean "expansion"?

Perhaps, if the work can be generalized to non-linear convolutional neural networks or networks that are trained and their dynamics, the findings would be new.

Minor comments:

1. I suggest not using a bold font in sentences.
2. I found the first paragraph very disconnected from the main focus of the paper. It's the authors' choice, but I recommend making a better connection or starting with the second paragraph. For example, does the paper directly explain why DNN fails in any of the problems in the first paragraph? No, but you may say indirectly they can provide insights.
3. Section 3, line 7, "kenel" --> "kernel".
4. There is an extra ")" in (11).

[1] Discrete-time signal processing by Oppenheim and Schafer.

**Summary Of The Paper:**

This paper provides the bottleneck of representations in linear convolutional networks. This is based on the frequency analysis of the input and how it propagates into the network. They characterize and analyze the network in detail. Here are some of their statements as contributions: Each frequency propagates independently of the other through network mapping. The convolution and other operations related to it (e.g., zero padding) damp high-frequency components. The last finding (in their own wording) is that upsampling results in strong signals repetitively appearing at certain frequencies.

**Summary Of The Review:**

I very much enjoyed reading the paper. The paper gives lots of insights into linear convolutional neural networks and how frequency information passes through the network. However, all those insights are already known in the literature. I do not recommend the acceptance of this work for the following three reasons:

1. Almost all the analysis and intuitions given by the paper are known for convolution operators (convolutions of convolutions is also a convolution). When the authors reduce their neural network into a convolution, then all results are immediate. Hence, results on the independence of frequency propagation, how down zero-padding works, and effect upsampling are already known for convolutions.
2. Assumptions and simplifications are extreme.
3. The paper does not acknowledge signal processing literature and its wording implies their work is new.

---

### Official Review · Reviewer_oTym · 2022-10-31

**Confidence:** 4
**Correctness:** 3
**Technical Novelty And Significance:** 2
**Empirical Novelty And Significance:** 2
**Recommendation:** 3

**Clarity, Quality, Novelty And Reproducibility:**

The organization of this paper is not very clear, with many theoretical findings intervened with bold-font bullets. The novelty of the theoretical results is limited. One should have no difficulty implementing the convolutional decoder network to verify the claims of this paper based on the description in the paper.

**Strength And Weaknesses:**

Strengths:

1. The authors performed a detailed study on the effect of different components in a decoder network on the spectrum domain by discrete Fourier transform.
2. Extensive experiments are conducted to support the claims of the authors.

Weakness:

1. The findings of the paper are not surprising, and using discrete Fourier transform to study the learning of neural networks on the frequency domain is not novel. Previous works have proved that neural networks tend to learn low-frequency components [1] with the help of discrete Fourier transforms [2,3]. More discussions need to be included on the significance of this paper compared to previous works.

2. The findings are limited to decoder networks. The theoretical proof does not consider common activation layers in decoder networks such as ReLU.

3. The paper's organization within Section 2 and Section 3 is messy.

Besides the above weakness, the other issue is about the (mathematical) novelty in many theoretical results. For example, Theorem 1, Corollary 1, and Theorem 2 are just another format of the very famous convolution theorem: Fourier transform of the convolution of two inputs signals is the elementwise product of the Fourier transform of the two input signals.

[1] Kiessling, Jonas, and Filip Thor. "A Computable Definition of the Spectral Bias." AAAI 2022.

[2] Rahaman, Nasim, et al. "On the spectral bias of neural networks." ICML 2019.

[3] Xu, Zhi-Qin John, et al. "Frequency principle: Fourier analysis sheds light on deep neural networks." arXiv preprint arXiv:1901.06523 (2019).



**Summary Of The Paper:**

This paper studies the effect of cascaded convolutional decoder network on the spectrum domain, and presents several findings summarized in the experiment section.

**Summary Of The Review:**

While this paper presents various effects on the spectrum domain of different components in a cascaded convolutional decoder network, the paper lacks mathematical novelty in its theoretical results and discussion/comparison with closely related works. The presentation of this paper should also be improved significantly.

---

### Decision · Program_Chairs · 2023-01-20

**Decision:**

Reject

**Justification For Why Not Higher Score:**

Reviewers noted significant issues with the paper's novelty and technical claims, and unanimously recommended rejection. We did not receive an author response to reviewer comments.

**Justification For Why Not Lower Score:**

N/A

**Metareview: Summary, Strengths And Weaknesses:**

The paper studies the properties of linear convolutional networks. It derives results showing how these networks act in the spectral domain, and argues that this tends to lead to a progressive attenuation of high frequencies. It also proves that upsampling introduces aliasing in the frequency domain. Experiments corroborate the main claims of the paper.

Reviewers expressed some appreciation for the paper's goal of studying feature propagation in the frequency domain. However, they noted several weaknesses, which led to a consensus to reject the paper. First, the results pertain only to linear decoding networks. Second, the main claims are of questionable novelty: the paper does not explain how they go beyond classical observations in signal processing (i.e., the convolution theorem and the frequency domain analysis of upsampling). Finally, reviewers felt that the paper did not sufficiently differentiate its claims from existing empirical work on frequency domain analysis of neural networks.